# Using integrated hydrological-hydraulic modelling and global data sources to analyse the February 2023 floods in the Umbeluzi catchment (Mozambique)

Luis Cea[1], Manuel Álvarez[1], Jerónimo Puertas[1]

[1]Universidade da Coruña, Water and Environmental Engineering Group, Center for Technological Innovation in Construction and Civil Engineering (CITEEC), Campus de Elviña, 15071 A Coruña, Spain

*Correspondence to*: Luis Cea (luis.cea@udc.es)

**Abstract.** On 9-13 February 2023 an intense flood event took place in the province of Maputo (Mozambique), resulting in severe damage to agricultural lands and transport infrastructure, and with serious consequences for the population. In the district of Boane, located a few kilometres downstream of the Pequenos Libombos dam, the flood destroyed many food crops as well as two bridges linking the district to Maputo, thus affecting the food security of the population. These events are quite frequent in this region, making necessary the delineation of improved flood hazard maps and the development of new flood risk management plans. We reproduce this flood event with a high resolution integrated hydrologic-hydraulic model fed with freely available global data sources, using a methodology that can be easily reproduced in other data-scarce regions. The model results are validated with observed estimations of the inflow to the Pequenos Libombos reservoir, with water marks left by the flood in the district of Boane, and with a Sentinel-1 image taken during the recession of the flood. We analyse the effect of the Pequenos Libombos reservoir on the flood hazard, which was subject to debate amongst the affected population and in the media. The results obtained show that integrated hydrologic-hydraulic models based on the two-dimensional shallow water equations, combined with global databases, are currently able to reasonably reproduce the extent and peak discharge of extreme flood events in data-scarce basins, and are therefore very useful tools for the development of flood management plans in these regions.

## 1 Introduction

As with many other African countries, Mozambique is highly exposed to the impact of floods and to the effects of climate change (Revilla-Romero et al. 2015; World Bank, 2019). This is mainly due to the high vulnerability of its communities, combined with the extreme rainfalls produced by the tropical storms and cyclones that occur on its coastline on average every two years (WMO, 2019). Moreover, Mozambique's population is forecast to grow from 30 million to 65 million over the next 30 years, and will concentrate near rivers, lakes and the coastline, thus increasing the exposure of these populations to the impact of floods. In light of this, Mozambique has made significant efforts in recent years to put in place flood risk evaluation and mitigation measures.

The heavy rains that occurred in southern Mozambique between 6 and 15 February 2023 resulted in local rainfall depths of 350 mm, causing widespread flooding and considerable damage, especially in the city of Maputo and its neighbourhood. According to the data provided by the National Institute for Disaster Risk Reduction and Management (INGD), as of 17 February, 43,426 people has been affected by the floods, with 16,600 people displaced and 10 deaths (OCHA, 2023). The district of Boane, located downstream of the Pequenos Libombos (PL) dam and crossed by the waters of the Umbeluzi river and its tributary Movene, was the most affected part of the province. With a very flat topography, many neighbourhoods in this area were wholly inundated and isolated, given that road traffic was interrupted on the EN2 National Road that connects the city of Matola to the village of Boane. The Mazambabine and Boane bridges were submerged and the drinking water treatment plant was disrupted, resulting in significant cuts in water supply to the population. Upstream of the PL dam, the steel

bridge over the Umbeluzi River linking Michangulene and Mafavuka settlements was swept away, and the bridge over the Calichane River on the EN3 National Road connecting Mozambique to eSwatini was partially destroyed.

The PL dam is located in the district of Namaacha, in the province of Maputo, and is the main water infrastructure located on the Umbeluzi River within the territory of Mozambique. It was built between 1981 and 1987, and its main purpose is the supply of water to the cities of Maputo, Matola and Boane, as well as the irrigation of agricultural lands. The storage capacity of the reservoir is 385 hm$^3$, corresponding to a maximum operation level or Normal Pool Level (NPL) of 47.00 m. The design flood level, or Maximum Pool Level (MPL), is at 49.55 m, just 0.5 m below the crest of the dam. The spillways of the dam are controlled by movable gates and have their crest at an elevation of 24.00 m, the reservoir's volume at that level being 8 hm$^3$. Thus, the reservoir's outlet discharge can be controlled for almost its full range of capacity.

During the first 8 days of February, the reservoir remained at an average level of 45.56 m, corresponding to a storage volume of 331.2 hm$^3$ and 86.1% of its NPL capacity. During the following two days, an intense rainfall event within the basin led to a maximum daily inflow discharge of 3,848 m$^3$ s$^{-1}$. The reservoir level reached 48.50 m, corresponding to 451.6 hm$^3$ and 117.3% of its NPL capacity. Under this critical situation, and in order to guarantee the structural safety of the dam, it was necessary to release water at the spillways' maximum capacity (around 2,870 m$^3$ s$^{-1}$). The situation remained practically unchanged until 10 February, when the reservoir level dropped slightly to 48.20 m.

In this study we reproduced the flood event that took place on 9-13 February 2023 in the Umbeluzi basin using the software Iber+, a GPU-enhanced high resolution integrated hydrologic-hydraulic model based on two-dimensional shallow water equations (Bladé et al. 2014; Cea and Bladé, 2015; García-Feal et al. 2018). This kind of modelling approach is becoming increasingly popular in recent years due to the development of efficient numerical solvers that implement different GPU or CPU parallelization techniques, making it possible to solve the two-dimensional shallow water equations in a whole catchment using grids of several millions of elements (Caviedes-Voullième et al. 2023; García-Feal et al. 2018; Morales-Hernández et al. 2021; Noh et al. 2018; Sanders and Schubert, 2019; Sharifian et al. 2023; Xia et al. 2019).

The model parameters were defined from standard non-calibrated values, and all input data were obtained from global data sources that are freely available worldwide, making the methodology reproducible anywhere. We evaluated the ability of this type of model to reproduce flood events in data-scarce regions, where it is necessary to rely on global databases, and where detailed observed data are not available for the calibration of model parameters. To this end, the model results were validated with: 1) inflows to the PL reservoir during the event, these provided by the regional water administration ARA-Sul, 2) maximum water depths, estimated from the identification of water marks left by the flood at different points in the district of Boane, and 3) the extent of the water, estimated from a Sentinel-1 image taken during the recession of the flood. Once the model was validated, the effect that the management of the PL reservoir had on the spatial extent of the inundation and in the maximum water depths reached in the surroundings of Maputo was analysed. For this purpose, three modelling scenarios were reproduced numerically: MS1 represents the actual management of the PL reservoir that took place during the storm event, MS2 reproduces what would have happened in the absence of the PL reservoir, and MS3 predicts what would have happened if the PL reservoir had been able to retain the total inflow hydrograph that arrived there during the storm event.

The results obtained show that integrated hydrologic-hydraulic models based on the two-dimensional shallow water equations, combined with global databases, are currently able to reproduce the extent and peak discharge of extreme flood events in data-scarce basins, and are therefore very useful tools for the development of flood management plans in these regions. The accuracy of the water depth predictions might however be limited in certain regions by the spatial resolution and accuracy of the global topographic data currently available. In the case of the February 2023 floods in the Umbeluzi catchment, it can be claimed that the management of the PL reservoir contributed to reduce the flood hazard in Boane, although the effect was relatively small due to the limited flood control capacity of the reservoir and the high magnitude of the flood. In the absence of the dam, the impact of the flood would have been far greater.

**2 Case study: The Umbeluzi catchment**

The Umbeluzi catchment is one of the largest in southern Mozambique. It has a total surface of 5,461 km$^2$, distributed between Mozambique (40.7 %), eSwatini (57.6 %), and South Africa (1.7 %). The Umbeluzi River originates in the foothills of the Malolotja Nature Reserve, at an altitude of 1,393 metres. After extending for about 290 km in a West-East direction, it flows into the Indian Ocean. The outlet of the catchment considered in this study (Figure 1) is located 7.2 km upstream of the junction of the Umbeluzi, Matola and Tembe rivers (coordinates UTM zone 36S; 443,993mE; 7,119,978mN) in the estuary of Espirito Santo.

The average elevation of the Umbeluzi basin is 346 m above mean sea level, ranging from 0 to 1,828 metres, with an average slope of 9.9%. The Central and Eastern part of the basin, which occupies 73.7% of its surface, has a flat or moderate slope (less than 12%). This part of the basin is divided by the Pequenos Libombos mountain range, which runs in a North-South direction along the border of Mozambique with eSwatini and South Africa. In this area the relief is undulating or very steep, with slopes over 75% and reaching maximum slope values close to 160%.

The average annual precipitation in the entire Umbeluzi basin during the period 1981-2010, estimated from the CHIRPS (Climate Hazards Group InfraRed Precipitation with Station data) daily data source (Funk et. al., 2015), was 744 mm. The spatial distribution of rainfall varies from 600 mm in the flat areas of the lower part of the basin, to 1,300 mm in the more mountainous headwater areas. The wet period is from October to March and accounts for 84 % of total annual precipitation.

For the purpose of this study, the Umbeluzi catchment was split in two complementary regions, as shown in Figure 1. The first region includes the catchment located upstream of the PL dam, while the second region includes the catchment located downstream of the PL dam. In what follows, we will refer to these two regions as U-PLD (Upstream Pequenos Libombos Dam) and D-PLD (Downstream Pequenos Libombos Dam).

The part of the basin located downstream of the PL dam (D-PLD in Figure 1) is one of the most flood-prone regions in the Maputo province. With an area of 1,723 km$^2$, this subbasin has been impacted by intense floods in the years 1966, 1972, 1977, 1984 and 2000. Also, during the hydrological years 2016 and 2020, the tropical cyclones Dineo, Chalane, Eloise and Guambe Eloise caused significant flooding. Historical records from a hydrometric station located in Boane, prior to the construction of the PL dam, indicate that the 1984 flood was the largest of all registered floods, with a maximum discharge in the order of 7,250 m$^3$ s$^{-1}$ (Lacamurima, 2003).

The analysis of flood hazard in this study focuses on an Area of Interest (AOI) of 313 km$^2$ distributed over the districts of Boane (83%) and Naamacha (17%) (Figure 1). This is a very vulnerable area in terms of flood damage since it is located just a few km downstream of the PL dam, thus receiving the outflow discharge from the reservoir, as well as the surface runoff generated by rainfall falling in the D-PLD subbasin, the major contribution of which is from the Movene river basin.

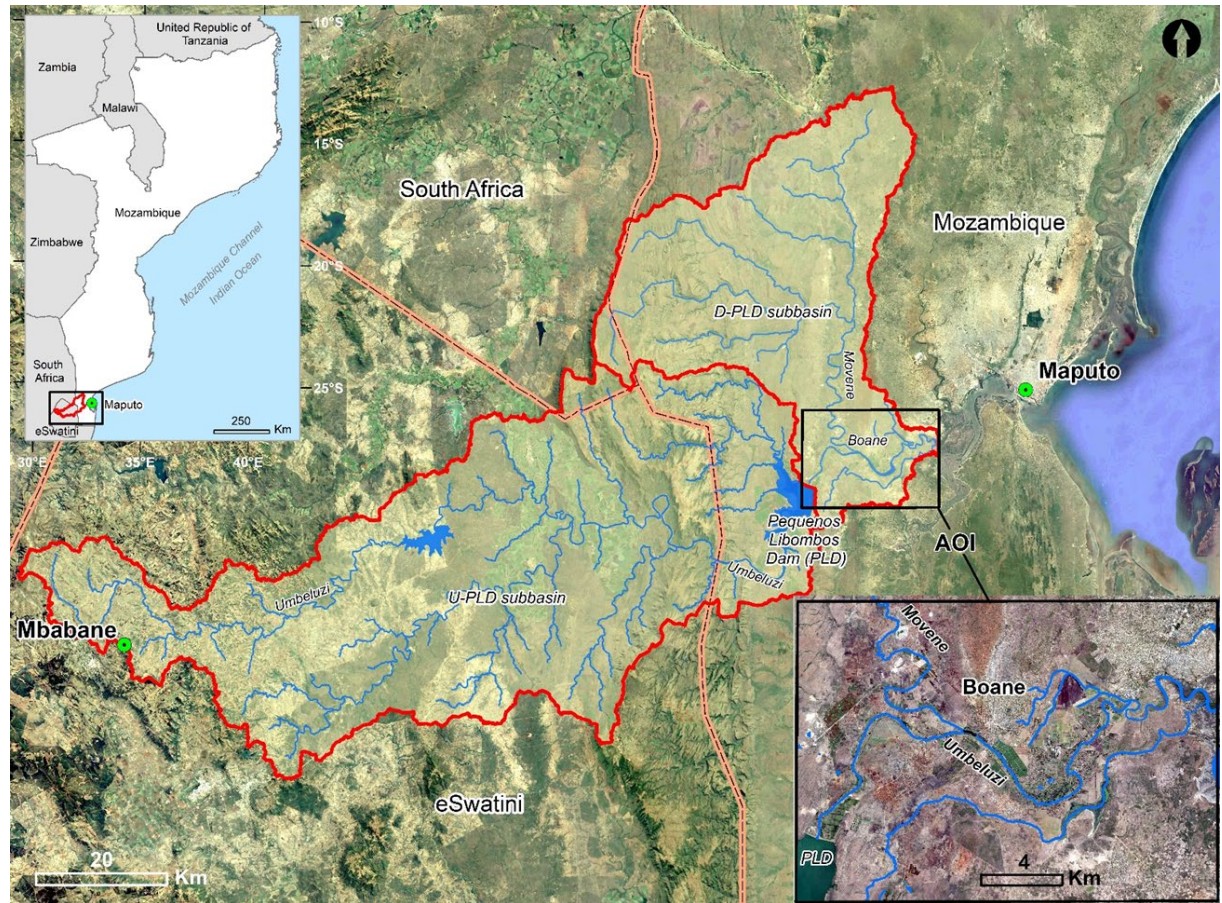

**Figure 1: Location map of the Umbeluzi catchment, divided in two subbasins, U-PLD and D-PLD, located respectively upstream and downstream of the PL dam, and definition of the Area of Interest (AOI) for this study. Background image © Google Earth V 7.3.6.9345, http://www.earth.google.com [13 April 2023].**

## 3 Methodology

### 3.1 Global data sources

The various global data sources used in this work are listed in Table 1, which also includes their spatial resolution and the URL where they can be retrieved without cost. In what follows, the most relevant features of each data set for hydrological modelling purposes are described.

| Variable | Data set | Resolution | Source |
|---|---|---|---|
| DEM | Copernicus GLO-30 | 30 m | https://panda.copernicus.eu/web/cds-catalogue |
| Rainfall | GPM IMERG Final Precipitation L3 | 10 km, 30 min | https://disc.gsfc.nasa.gov |
| Land Cover | ESA WorldCover 10m 2021 v200 | 10 m | http://due.esrin.esa.int/page_globcover.php |
| Infiltration | GCN250 | 250 m | https://doi.org/10.6084/m9.figshare.7756202.v1 |

**Table 1: Data sets used in the numerical model.**

#### 3.1.1 Digital Elevation Model

The topography of the whole Umbeluzi catchment was obtained from the Copernicus GLO-30 Digital Elevation Model (DEM), which has a spatial resolution of 1 arc-sec (roughly 30 m). This is the highest resolution global DEM generated and provided free of charge by the European Space Agency (ESA). It was obtained by the ESA after processing the data from the TanDEM-X mission, which took place between 2011 and 2015 (Krieger et al., 2013; Zink et al. 2021), covering the whole Earth with a

spatial resolution of 12 m. The ESA also provides a DEM obtained from TanDEM-X at a spatial resolution of 0.4 arc-sec (EEA-10), but only covering European states. There is also a commercial version of TanDEM-X with a spatial resolution of 12 m (WorldDEM$^{TM}$) edited by Airbus Defence and Space (Bayburt et al., 2017), but it is not available free of charge.

Several recent studies in various regions of the world concluded that Copernicus GLO-30 is the DEM with the best overall performance for hydrological modelling purposes that is currently available for free (Cea et al. 2022; Garrote, 2022; Guth and Geoffroy, 2021; Maresova et al., 2021), producing a better representation of the terrain for hydrological and hydraulic computations than other commonly used 1 arc-sec DEMs, such as ALOS, ASTER, NASADEM and SRTM v3.

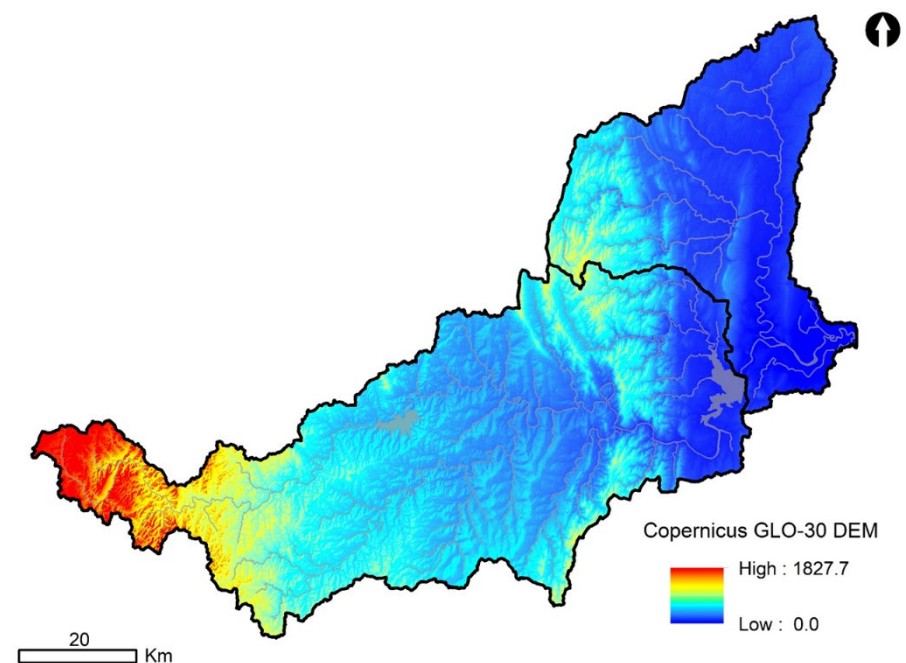

Figure 2: Topography of the Umbeluzi catchment obtained from Copernicus GLO-30 DEM.

### 3.1.2 Rainfall

There are currently several freely available data sets providing global rainfall estimates that can be used for a variety of hydrological studies. In the case of flood modelling, the products derived from the Global Precipitation Measurement (GPM) satellite constellation of the National Aeronautics and Space Administration (NASA) (Huffman et al., 2020) are of particular interest due to their low latency and high spatial and temporal resolution. The Integrated Multi-satellitE Retrievals for GPM (GPM-IMERG) provides precipitation estimates since March 2014 within the $60^{o}$ N–S latitude band, at maximum spatial and temporal resolutions of $0.1^{o}$ (roughly 10 km) and 30 minutes, with latencies of 4 h (Early Run), 14 h (Late Run) and 3-4 months (Final Run). When modelling relatively short and intense rainfall events, having rainfall estimates every 30 minutes is a clear advantage over other global products that work at lower temporal resolutions, as for instance CHIRPS, which provides estimates with a resolution of 1 day. Pradhan et al. (2022) provide a review of different GPM-IMERG validation studies at various locations around the globe. Other studies as Saouabe et al. (2020), Liu et al. (2020) or Tapiador et al. (2019), among others, also evaluated the accuracy of GPM-IMERG compared to observed rainfall in different parts of the world, concluding that the use of GPM contributes extraordinarily to improve the monitoring of extreme rainfall events when ground-based precipitation data is not available. In the case of Africa, a recent review by Gosset et al. (2023) about the role of satellite observations for monitoring pluvial and fluvial flood highlighted that major recent flood events in Africa have been well depicted by satellite observations, illustrating the feasibility of satellite monitoring for better surveillance of the food risk in this region.

In the present study we have used the GPM-IMERG Late Run data set, with $0.1^{o}$ and 30 minutes spatial and temporal resolutions, to represent the rainfall fields during the storm event that took place on 7-14 February 2023 over the Umbeluzi catchment. The Umbeluzi basin is covered by 66 rainfall pixels. The rainfall estimates between 6 February 2023 at 00:00 and

155 15 February 2023 at 00:00 (432 files) were used as rainfall input in the numerical model. Figure 3 shows the spatial distribution of rainfall depth over the catchment during the simulation period, as well as the basin-averaged rainfall depth-duration curve. Most of the rainfall (around 170 mm) fell on 8-9 February, while a second burst of about 50 mm fell within 10 hours in 12 February. The total basin-averaged rainfall depth during the event was around 240 mm.

The analysis of the spatial pattern of the rainfall over the basin shows that there was a cluster with cumulative rainfall depths
above 300 mm covering an area of 1,084 km². Within this core, a cell with a local maximum of 356 mm was located 12 km west of the PL dam. The spatial distribution of the rainfall event also shows that, towards the upper part of the basin, the cumulative rainfall depth decreases progressively until reaching a local minimum of 79 mm. This distribution pattern, with maxima close to the outlet of the U-PLD subbasin, contributed to reducing the response time of the basin, thus increasing the peak discharges flowing into the PL dam.

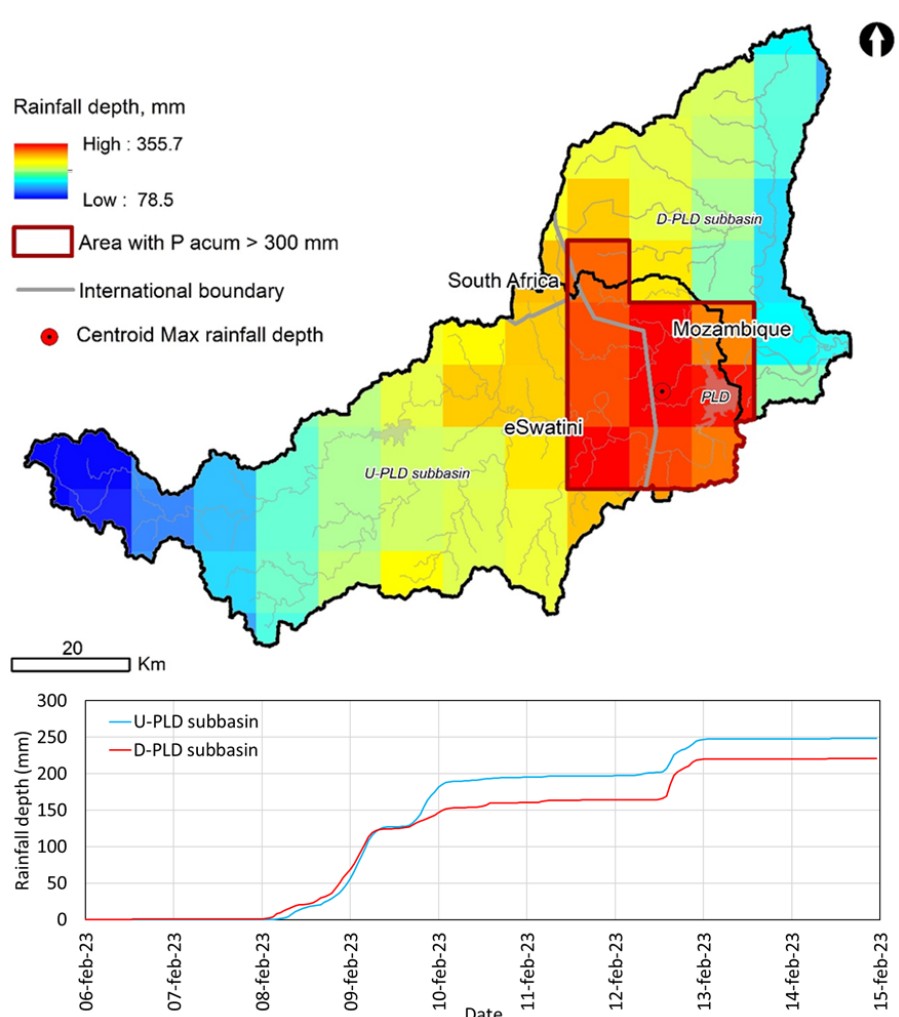

**Figure 3: Spatial distribution of rainfall depth the Umbeluzi catchment from 7-14 February 2023, obtained from the GPM-IMERG database, and time series of basin-averaged rainfall depth in the U-PLD and D-PLD subbasins.**

### 3.1.3 Land cover

The propagation of overland flow over the hillslopes and along the river network was computed with the software Iber, which solves the two-dimensional shallow water equations, using a roughness coefficient that depends on the land cover to characterise the bed friction between the terrain and the water. Hence, land cover maps are needed in order to estimate the spatial distribution of the roughness coefficient across the whole catchment.

From the various land cover maps that are currently available at the global scale, we have used WorldCover 10m 2021, recently
released by ESA. This product was generated within the framework of the ESA WorldCover project, itself part of the 5th Earth

Observation Envelope Programme (EOEP-5). It provides a global land cover classification for 2021 at a spatial resolution of 10 m, derived from Sentinel-1 and Sentinel-2 data, including 11 land cover classes. It is currently the most recent and highest resolution global land cover product available free of charge. The land cover distribution across the Umbeluzi catchment is shown in Figure 4. Trees and grassland are the two predominant land uses, each making up around 38% of the basin. Cropland and shrubland cover 12% and 10% of the catchment surface respectively, while the other 4 land uses present in the basin (built-up, sparse vegetation, permanent water bodies and herbaceous wetlands) represent only 2% of the surface.

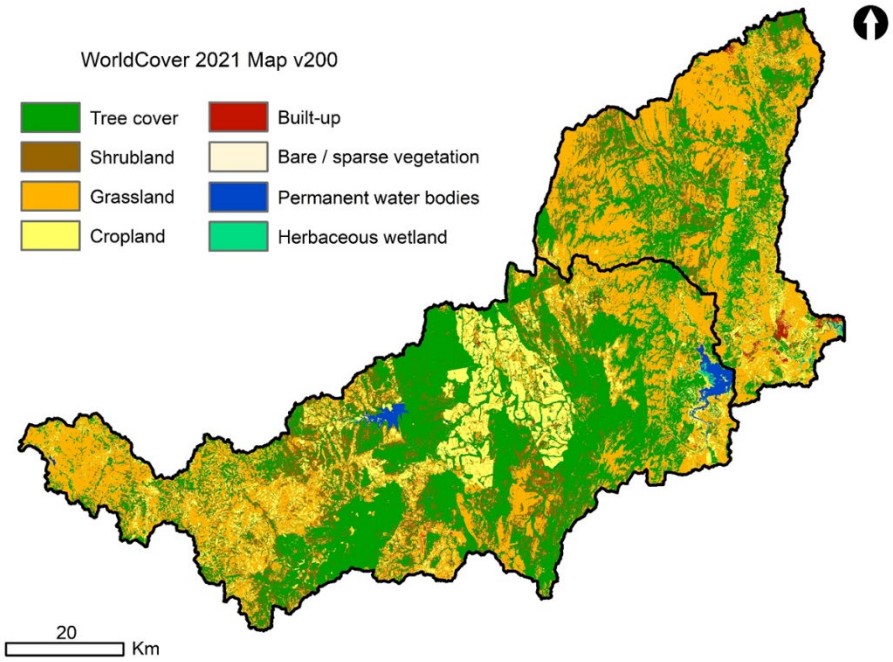

**Figure 4: Land cover in the Umbeluzi catchment obtained from ESA WorldCover 10m 2021 v200.**

### 3.1.4 Infiltration

The infiltration capacity of the terrain can be characterised using different empirical formulations (Singh, 2017). In the case of flood modelling, one of the methods most commonly used to estimate the potential infiltration of the soil is the Natural Resources Conservation Service (formerly Soil Conservation Service) Curve Number (SCS-CN). In the absence of detailed field data, it has the advantage of using a single parameter (CN) that has been extensively tabulated as a function of the hydrological soil group, the land cover, and the terrain slope (Singh, 2017). Thus, it is especially suitable in data-scarce regions, where the lack of field data that could be used for calibration hinders the application of other (also well-known) formulations such as those of Green-Ampt and Horton.

In order to estimate the potential infiltration in the numerical model, we have used the data set GCN250 (Jaafar et al., 2019), which includes a global estimation of CN for the whole Earth with a spatial resolution of 250 m. The CN values provided by this data set were derived by considering global maps of hydrological soil groups (HYSOG250m) and land cover (ESA CCI Land Cover project). The HYSOGs250m data set (Ross et al., 2018) was specially derived in order to support SCS-CN runoff modelling at global scales using soil data from the FAO Harmonized World Soil Database.

The data set includes CN estimates for average, wet and dry antecedent moisture content (AMC) of the soil, computed according to the SCS-CN methodology. In order to model the flood of 9-13 February 2023, we considered wet AMC conditions, since the 5-day antecedent rainfall depth in the basin was slightly greater than 50 mm.

Figure 5 shows the spatial distribution of CN within the Umbeluzi basin for wet AMC soil conditions, which varies between 85 and 98 in the hillslopes and floodplains, with a basin-averaged value of 91.

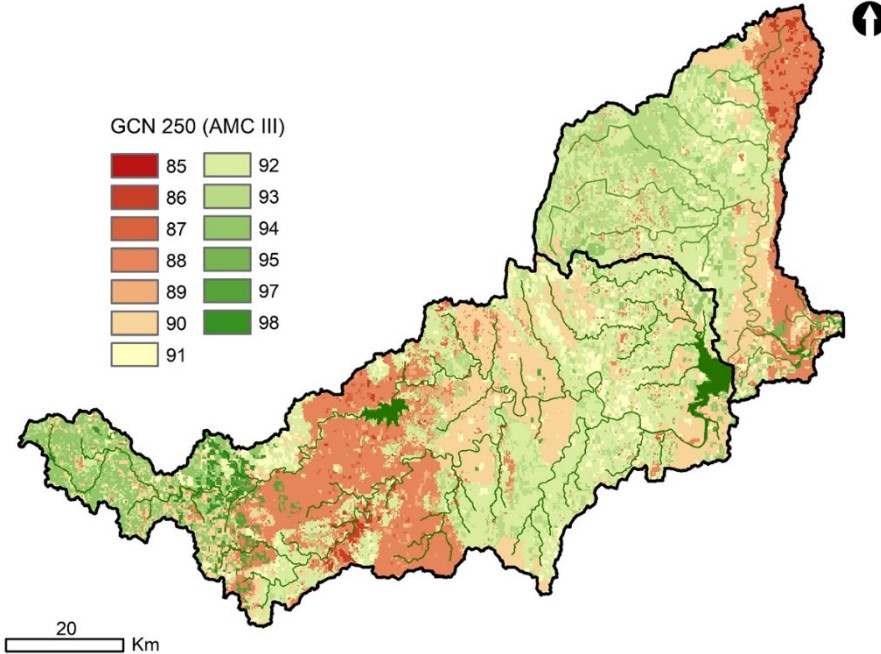

**Figure 5: Spatial distribution of CN for wet AMC conditions in the Umbeluzi catchment, obtained from GCN250 (Jaafar et al., 2019).**

## 3.2 Numerical model

The software used in this study (Iber) solves the 2D shallow water equations (2D-SWE) in the whole catchment with an integrated hydrological-hydraulic modelling approach. Iber implements rainfall and infiltration processes within the shallow water equations, allowing the simulation of rainfall-runoff transformation and river inundation processes simultaneously (Cea and Bladé, 2015). The model implements an unstructured finite volume solver for the 2D-SWE, allowing the user to adapt the mesh to the basin morphology, and to define a variable mesh size in different regions of the study area. This kind of modelling approach has been used in previous studies, in which its suitability for modelling rainfall-runoff transformation and overland flow propagation at the catchment scale during intense flood events has been shown (Cea et al. 2022; García-Alén et al. 2022; Sanz-Ramos et al., 2021; Moral-Erencia et al. 2021; Xia et al. 2019). Moreover, the application and validation of Iber to event-based hydrological computations at different spatial scales has been presented in several previous studies (Cea et al., 2010; Fraga et al., 2019; García-Alén et al. 2023; Sanz-Ramos et al., 2018; Sanz-Ramos et al., 2021; Tamagnone et al., 2020; Uber et al., 2021). The High Performance Computing (HPC) implementation of the software Iber (Iber+) is especially suitable for integrated hydrological-hydraulic modelling applications, because it can achieve speed-ups of two orders of magnitude with respect to the standard sequential implementation (García-Feal et al. 2018).

A different model was built for the U-PLD and D-PLD subbasins, as shown in Figure 6. The U-PLD model was used to compute the hydrograph entering the PL reservoir, and to validate the modelling approach by comparing the computed hydrograph with the observed one. The D-PLD model was used to compute the flood hazard in the district of Boane, considering in an integrated way the reservoir's outlet hydrograph during the event and the overland flow generated by the rainfall falling directly in the D-PLD subbasin.

In both models the spatial domain was discretised with an unstructured mesh of triangular elements, this adapted to the basin's morphology, introducing an explicit representation of the river network and using different element sizes in the hillslopes and in the river streams. To this end, the river network was defined from the DEM, considering first a minimum contributing drainage area (CDA) of 1 km², and then keeping only the streams with a Strahler order larger than 5. Once the stream network was delineated in this way, the width of the main channel was estimated to 100 m, based on visual inspection of ortophotos. The size of the mesh elements ranged from 25 m in the main river reaches to 80 m in the hillslopes. It should be noted that the distinction between the river network and hillslopes obtained in this way is only relevant in order to define the mesh size and

the bed roughness coefficient, and does not have any implications in terms of the type of equations or the numerical schemes implemented to compute the propagation of runoff, since the same numerical solver is applied to the whole spatial domain. This type of discretisation of the spatial domain has already been applied in previous studies with good results (Cea et al., 2022; Komi et al., 2017; Uber et al., 2021).

When applying this procedure to the Umbeluzi catchment, the river networks obtained have total lengths of 530 km and 350 km in the U-PLD and D-PLD models respectively (Figure 6). Those streams were discretised with 120k elements in the U-PLD model and 83k elements in the D-PLD model. Regarding the hillslopes, in the U-PLD model they covered a surface of 3,681 km² and were discretised with over 1.4 million elements, while the D-PLD model had around 1.1 million elements to cover a hillslope surface of 1,697 km². Considering both models and the whole Umbeluzi catchment, the total modelled surface

was 5,461 km², and the total number of elements was approximately 2.6 million.

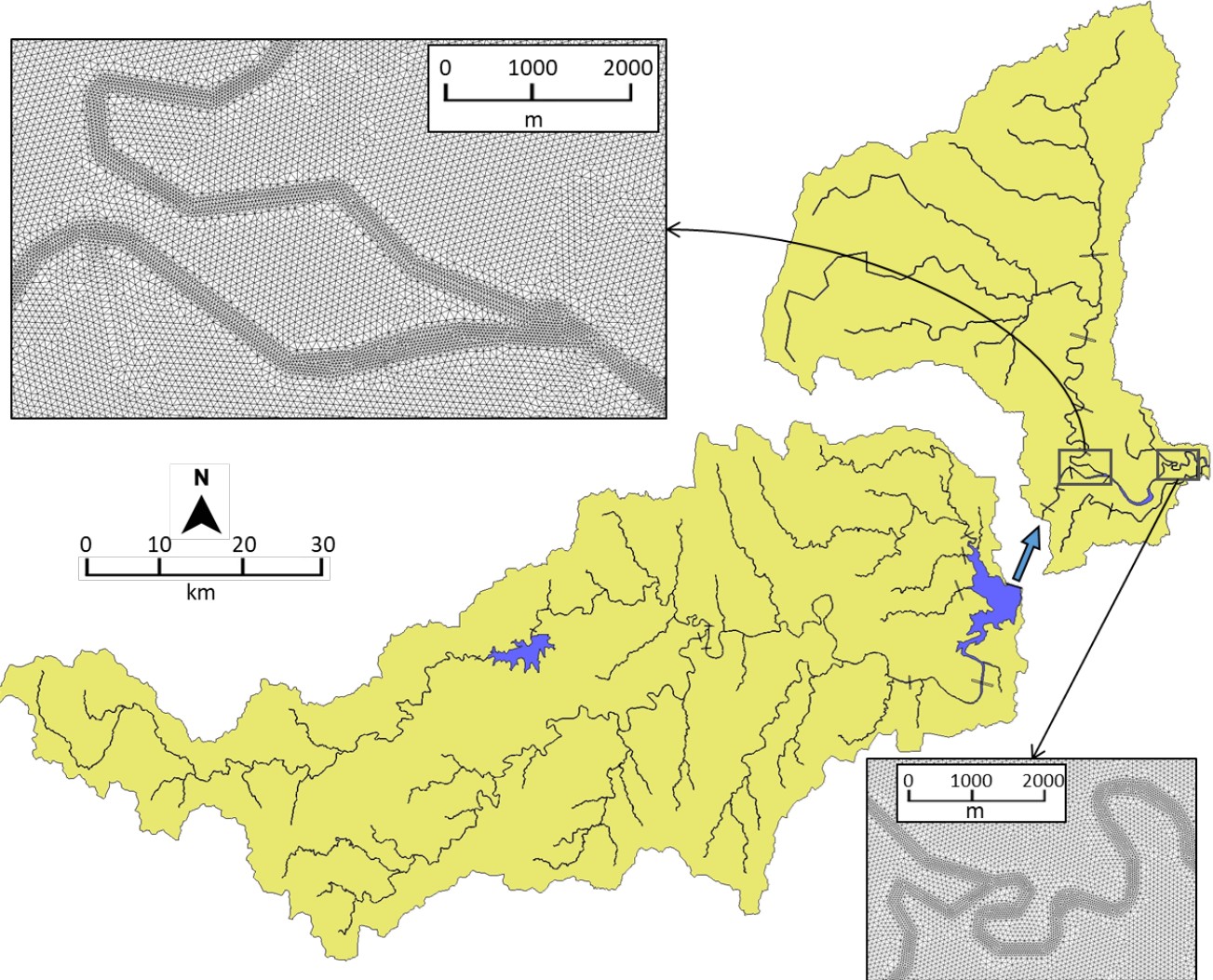

**Figure 6: Implementation in Iber of the Umbeluzi catchment. Geometry of the U-PLD and D-PLD models and details of the numerical discretisation.**


The hydrodynamic equations solved by the software Iber can be expressed as:

$$\frac{\partial h}{\partial t} + \frac{\partial q_x}{\partial x} + \frac{\partial q_y}{\partial y} = R - i \tag{1}$$

$$\frac{\partial q_x}{\partial t} + \frac{\partial}{\partial x}\left(\frac{q_x^2}{h} + g\frac{h^2}{2}\right) + \frac{\partial}{\partial y}\left(\frac{q_x q_y}{h}\right) = -gh\frac{\partial z_b}{\partial x} - g\frac{n^2}{h^{7/3}}|q|q_x \tag{2}$$

$$\frac{\partial q_y}{\partial t} + \frac{\partial}{\partial x}\left(\frac{q_x q_y}{h}\right) + \frac{\partial}{\partial y}\left(\frac{q_y^2}{h} + g\frac{h^2}{2}\right) = -gh\frac{\partial z_b}{\partial y} - g\frac{n^2}{h^{7/3}}|q|q_y \qquad (3)$$

where $h$ is the water depth, $q_x$, $q_y$ and $|q|$ are the two components of the unit discharge and its modulus, $z_b$ is the bed elevation, $n$ is the Manning coefficient, $g$ is the gravity acceleration, $R$ is the rainfall intensity, and $i$ is the infiltration rate. All the input data and parameters (rainfall fields, infiltration parameters and Manning coefficient) can vary in space.

As noted in Sect. 3.1.4, the soil infiltration capacity was modelled using the SCS-CN method. The spatial distribution of CN obtained from the GCN250 data set (Jaafar et al. 2019) was used to assign the CN in the hillslopes and floodplains, but its spatial resolution is not high enough to capture in detail the river network or to define precisely the water bodies, such as the reservoirs. Therefore, the value of CN in the river network and reservoirs was imposed manually to 100 in order to force the infiltration to zero in these areas.

The Manning's coefficient was defined by considering eight different land covers: Tree cover, Shrubland, Grassland, Cropland, Built-up areas, Bare/Sparse vegetation, Herbaceous wetland and Permanent water bodies. Their spatial distribution with a resolution of 10 m was obtained from the ESA WorldCover 10m 2021 v200 land cover map (Figure 4), except for the Permanent water bodies, which were assigned manually in order to achieve a more precise definition of their geometry than ESA WorldCover 10m 2021 v200. The Manning coefficients assigned to each land use are shown in Table 2.

| Land cover | Manning (s.m$^{-1/3}$) |
|---|---|
| Tree cover | 0.070 |
| Shrubland | 0.060 |
| Grassland | 0.035 |
| Cropland | 0.050 |
| Built-up areas | 0.100 |
| Bare/Sparse vegetation | 0.030 |
| Herbaceous wetland | 0.040 |
| Permanent water bodies | 0.025 |

**Table 2: Manning coefficient assigned to each land cover considered in ESA WorldCover 10m 2021 v200.**

Rainfall intensity fields were defined in the model with spatial and temporal resolutions of 10 km and 30 minutes respectively, using raster files obtained directly from the GPM-IMERG data base (Sect. 3.1.2). The numerical simulation extended from 6 February 2023 at 00:00 to 15 February 2023 at 00:00 (9 days).

Regarding boundary conditions, in the U-PLD model only one outlet boundary was defined at the dam location, where the

water surface elevation of the reservoir was imposed as a constant value during the whole simulation. As for the D-PLD model, inlet and outlet boundaries were defined respectively at the dam location and at the catchment outlet. At the inlet boundary, different inflow hydrographs were imposed for each modelling scenario, as will be described in Sect. 3.4, below. At the outlet boundary a supercritical flow condition was imposed, after verification that this condition did not affect the results of the flood extent in Boane.

The model was run on a standard desktop with a NVIDIA GeForce RTX 3080 Ti, which is an affordable GPU. With this hardware configuration, the simulations in the U-PLD and D-PLD models took around 25 and 15 minutes of computational time to reproduce the whole period of 9 days (i.e. each model runs about 600 times faster than real time).

### 3.3 Data for model validation

#### 3.3.1 Inflow and outflow discharges from the PL reservoir

The inflow and outflow daily discharges from the PL reservoir from 6 February to 15 February 2023, as well as the water surface elevation in the reservoir, were provided by ARA-Sul, and are shown in Figure 7. While the outflow hydrograph was controlled by ARA-Sul through the operation of the dam spillways and outlets during the event, the inflow hydrograph was derived from a daily mass balance in the reservoir, considering the controlled outflow and the daily evolution of its water

surface elevation. Each discharge value in Figure 7 was computed daily at 07:00 of the current day, and corresponds to the average discharge over the previous 24 hours, while the water surface elevation values correspond to the actual water level in the reservoir at 07:00 of the current day.

At the beginning of the event the reservoir was almost full. The water surface elevation was 45.76 m (corresponding to a storage volume of 336 hm³ and 87% of its storage capacity), very near to its Normal Pool Level (NPL), which is 47.00 m and 385 hm³. Thus, when the inflow discharge began to raise markedly, on 9 February 2023, the reservoir spillways were opened to their maximum capacity (circa 2,850 m³ s⁻¹), releasing around 250 hm³ per day (i.e. around 65% of the reservoir's storage capacity). This situation was maintained for 2 days, until the inflow discharge began to decrease. Over these two days the daily average inflow discharge was 3,900 m³ s⁻¹ and 2,700 m³ s⁻¹ respectively, leading to an increase in the reservoir's water level to 48.50 m, i.e. 1.5 m above its NPL and just 1.05 m below its Maximum Pool Level (MPL) for the design flood. At the end of the event, the water storage in the reservoir was similar to the initial level, corresponding to a water surface elevation of roughly 46 m.

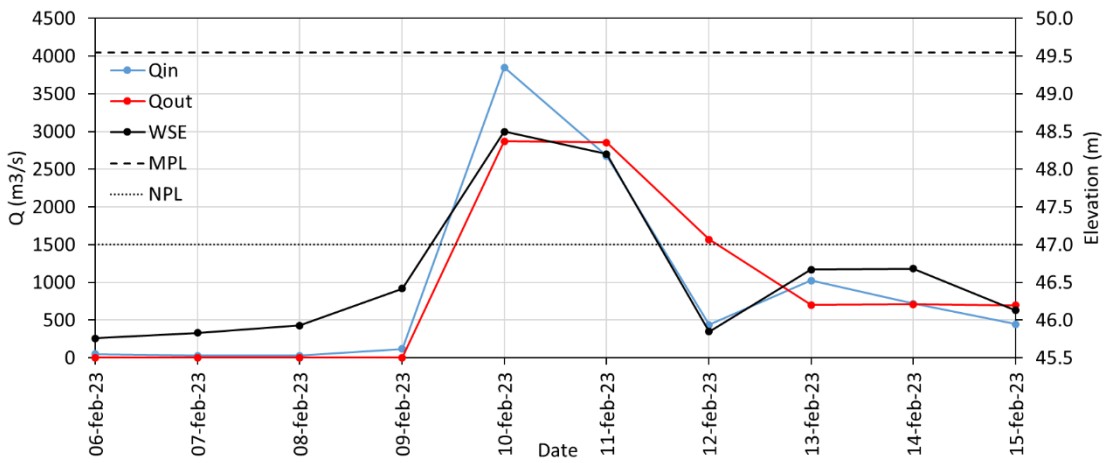

**Figure 7. Daily inflow (Qin), daily outflow (Qout), and water surface elevation (WSE) in the PL reservoir during the flood event. The Normal and Maximum Pool Levels (NPL and MPL) are also provided for reference. Daily discharges correspond to the average over the previous 24 hours.**

### 3.3.2 Maximum water depths

On 20-21 March 2023, the neighbourhoods of Boane district most damaged by the flood were visited by technicians from the regional water administration ARA-Sul, in order to estimate the level reached by the waters during the flood event. During this field work, inundation marks on buildings and other infrastructure were identified. At each point identified, the maximum water depth reached during the flood was estimated as the difference between the elevation of the inundation mark and the terrain at that location. A total of 20 water marks were thus identified; their locations are shown in Figure 8.

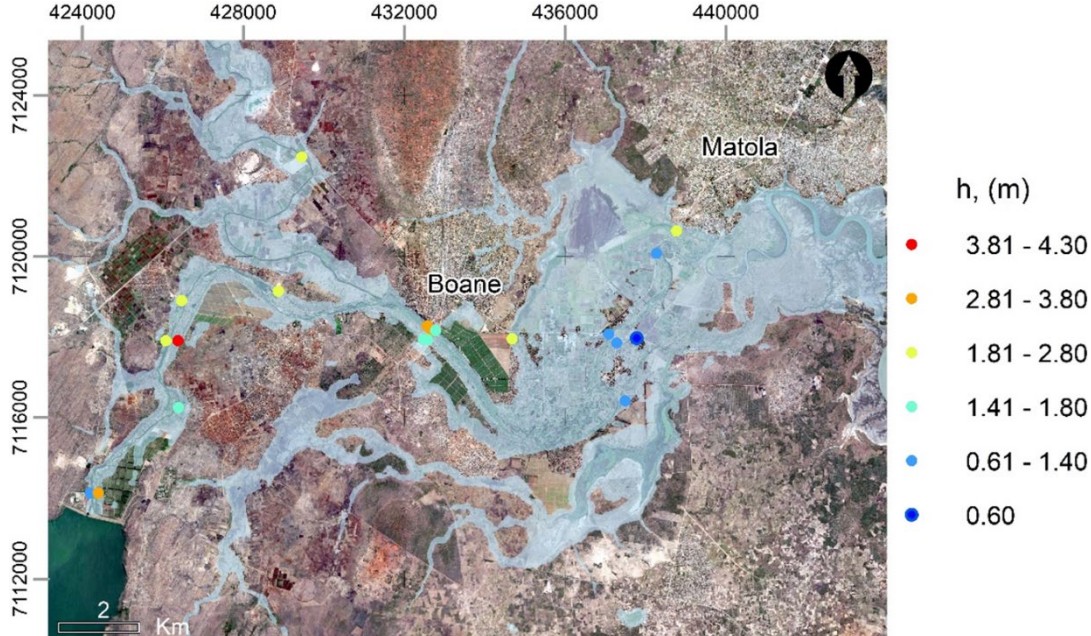

**Figure 8: Location of the points at which the maximum water depth during the flood was estimated from field observations of the marks left by the water. The color of the dots indicates the maximum water depth estimated. The shaded area represents the maximum flood extent for the scenario MS1. Background image © Google Earth V 7.3.6.9345, http://www.earth.google.com [13 April 2023].**

| ID | X (m) | Y (m) | h field (m) | Zb DEM (m) |
|----|-------|-------|-------------|------------|
| 1 | 424,197.7 | 7,114,128.6 | 1.3 | 28.3 |
| 2 | 424,397.7 | 7,114,129.8 | 3.7 | 23.1 |
| 3 | 426,076.4 | 7,117,905.1 | 2.6 | 17.9 |
| 4 | 426,471.0 | 7,118,904.1 | 2.5 | 16.4 |
| 5 | 428,870.8 | 7,119,138.9 | 2.6 | 15.6 |
| 6 | 429,453.1 | 7,122,464.6 | 2.8 | 10.3 |
| 7 | 432,577.0 | 7,118,272.5 | 3.8 | 8.5 |
| 8 | 432,677.6 | 7,118,162.3 | 3.4 | 7.2 |
| 9 | 432,777.6 | 7,118,162.8 | 1.6 | 11.3 |
| 10 | 432,478.7 | 7,117,939.8 | 1.8 | 12.6 |
| 11 | 432,578.7 | 7,117,940.3 | 1.6 | 10.6 |
| 12 | 426,385.9 | 7,116,245.5 | 1.8 | 18.4 |
| 13 | 426,376.5 | 7,117,906.8 | 4.3 | 17.1 |
| 14 | 434,679.4 | 7,117,951.0 | 2.3 | 6.9 |
| 15 | 437,079.7 | 7,118,073.5 | 1.4 | 7.1 |
| 16 | 437,280.8 | 7,117,853.0 | 1.1 | 9.3 |
| 17 | 437,487.8 | 7,116,414.2 | 1.3 | 8.2 |
| 18 | 437,780.5 | 7,117,966.1 | 0.6 | 6.7 |
| 19 | 438,270.7 | 7,120,072.7 | 1.2 | 7.1 |
| 20 | 438,768.3 | 7,120,628.8 | 2.4 | 3.0 |

**Table 3: Maximum water depths during the flood, estimated from field observations of water marks at several locations in the Boane district. The terrain elevation retrieved from the Copernicus GLO-30 DEM is also shown. X and Y coordinates correspond to UTM zone 36S.**

### 3.3.3. Water extent estimated from Sentinel-1 data

The ESA satellite Sentinel-1A captures Earth images with a 12-day repeat cycle. It is equipped with a C-band Advanced Synthetic Aperture Radar (SAR) that enables it to capture images under cloudy or rainy weather conditions, during both day and night, and hence is very suitable for estimating the extent of the water surface (Nemni et al., 2020). In Sentinel-1A images,

water can be distinguished from the surrounding terrain because its different roughness causes a different level of backscatter (Kuntla and Manjusree, 2020).

During the storm event of February 2023, the only available image from Sentinel-1 covering the AOI was taken on 14 February 2023 at 03:20 UTC, and it was used in this study to estimate the water extent with a pixel resolution of 10 m. The Sentinel Application Platform (SNAP) Toolkit developed by ESA for processing SAR-C images (Zuhlke et al., 2015) was used for this purpose. A pixel-based comparison with the water extent obtained with Iber at the same time was carried out using the following performance indices (Bennett et al., 2013; Bermudez et al., 2019; Costabile et al., 2020; Grimaldi et al., 2016):

$$HR = \frac{TP}{TP + FN} \qquad\qquad FAR = \frac{FP}{TP + FP} \qquad\qquad CSI = \frac{TP}{TP + FP + FN} \qquad (4)$$

where HR is the Hit Rate (proportion of the area observed as flooded in the satellite image that the model also predicts as flooded), FAR is the False Alarm Ratio (proportion of the area predicted as flooded by the model that has been observed as dry in the satellite image), CSI is the Critical Success Index, TP are the true positives (number of grid cells correctly predicted as flooded), FP are the false positives (number of cells that the model predicted as flooded but were observed as dry), and FN are the false negatives (number of cells predicted as dry but observed as flooded). The three ratios vary between 0 and 1. The

HR penalises underprediction and its optimal value is 1, meaning that all the area observed as flooded is correctly identified. On the other hand, the FAR penalises overprediction and its optimal value is 0, meaning that all the area predicted as flooded by the model is also identified as flooded by the satellite image. The CSI penalises both, overprediction and underprediction. Thus, to have a CSI close to 1 (its optimal value) the model prediction must match the satellite observation. As the model overpredicts or underpredicts the observations, the value of the CSI will diminish towards 0.

**3.4 Modelling scenarios**

The total discharge arriving at Boane originates from both the PL reservoir and the unregulated D-PLD subbasin. The reservoir's outflow hydrograph can, to a certain degree, be controlled by its management strategy, but this is not the case with the runoff originating from the D-PLD subbasin, in that there is not any regulation structure there. In order to better understand and quantify the contributions of both sources of flooding to the total hydrograph arriving at Boane, three flooding scenarios

were reproduced with the numerical model (Table 4).

MS1 reproduces the flood event that took place on February 2023, considering the actual management of the PL reservoir on those dates. The purpose of this scenario is to validate the model predictions and to reproduce the flood extent and depths in Boane during the event. Thus, the actual outflow discharge from the PL reservoir (Figure 7) was imposed as an inlet discharge at the location of the PL dam in the D-PLD model.

MS2 is a prediction of what would have happened if the PL reservoir had not existed. The aim here is to quantify the effective flood control exerted by the PL dam, by comparing the MS2 results with those of MS1 in terms of flood hazard in Boane. In this case, the inlet discharge at the location of the PL dam in the D-PLD model is equal to the outflow discharge computed in the U-PLD model, as schematised in Figure 6.

Finally, MS3 predicts what would have happened if the PL reservoir had been able to retain and control the total inflow arriving

there during the storm event. As noted in Sect. 3.3.1, this is not possible with the dam itself, since the volume of water arriving at the reservoir between 11-14 February 2023 roughly doubled its maximum capacity. The aim here is to quantify the maximum reduction in flood hazard that could have been achieved in Boane if a far larger reservoir than PL had existed. In this scenario, the inlet discharge in the D-PLD model is null, and the only source of flooding is the runoff generated in the D-PLD subbasin, the greatest contribution to which is from the Movene river basin (Figure 1).

| Modelling Scenario | Brief description | Purpose | Forcing in the D-PLD model |
|---|---|---|---|

| | | | |
|---|---|---|---|
| MS1 | Actual flood event | Validate the model | Rainfall + Regulated outflow from the PL dam |
| MS2 | No PL reservoir | Comparison with MS1 to quantify the flood control exerted by PL dam | Rainfall + Natural outflow from the U-PLD model |
| MS3 | Reservoir with unlimited storage | Quantify the maximum reduction in flood hazard that could be achieved with a far larger reservoir than PL | Rainfall |

**Table 4: Description of the modelling scenarios analysed.**

## 4 Results and discussion

### 4.1 Model validation. Scenario MS1

The ability of the proposed modelling approach to reproducing the flood event that took place on 9-13 February 2023 in the Umbeluzi catchment was assessed by comparing the predicted and observed values of: 1) the inlet hydrograph to the PL reservoir, 2) the maximum depths reached by the water at the locations shown in Table 3, and 3) the extension of the water in Boane at the time when the available Sentinel-1 image was taken (14 February 2023 at 03:20 UTC).

#### 4.1.1 Hydrographs into the PL reservoir

Figure 9 shows the observed and computed daily average discharge during the flood event. The numerical agreement with the observations is very good, with a coefficient of determination of $R^2 = 0.96$, and no relevant trend in the results. Therefore, the model parametrisation, and more specifically the assumption of wet AMC conditions in order to compute the infiltration losses with the SCS-CN method, can be considered as a plausible hypothesis that effectively represents the response of the catchment during the flood event of February 2023.

The hourly discharge into the PL reservoir computed with Iber, which is also represented in Figure 9, shows that the peak hourly discharge flowing into the reservoir (5,700 m$^3$ s$^{-1}$) was almost 50% higher than the maximum daily discharge (3,780 m$^3$ s$^{-1}$).

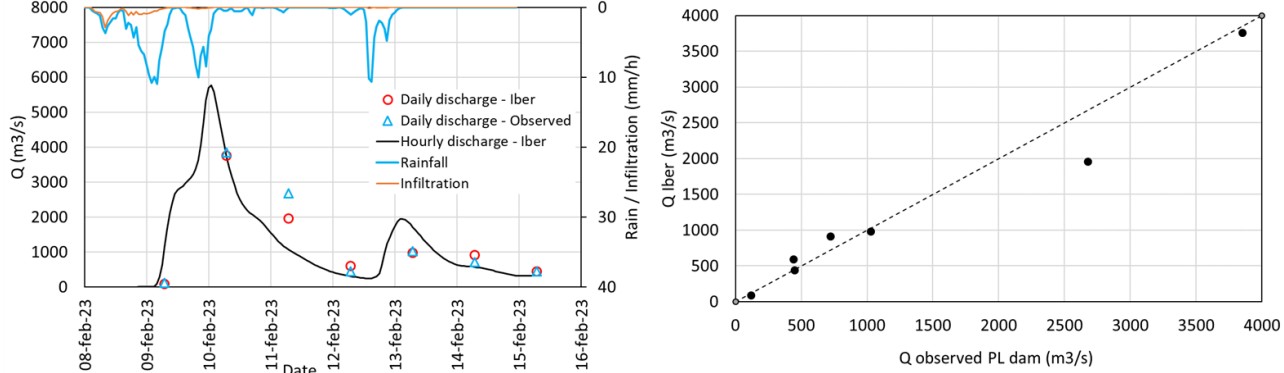

**Figure 9: Observed and modelled inlet discharges into the PL reservoir. Time series of daily and hourly discharge (left), simulated hourly discharge (top-right), and observed vs. simulated average daily discharge showing the identity line (right).**

#### 4.1.2 Maximum water depths

The most vulnerable areas in terms of flood damage are the floodplains located between the PL dam and the surroundings of Maputo, and thus it is in this area where the analysis of flood extent and maximum water depths was focused. This area of interest, indicated as AOI in Figure 10, receives the outflow discharge from the PL reservoir, as well as the surface runoff generated by the rainfall falling in the D-PLD subbasin, shown as a shaded area in Figure 10. Figure 10 also shows the hydrographs computed at three river cross-sections located within the AOI, around the confluence of the rivers Umbeluzi and Movene.

The outflow hydrograph from the reservoir is barely transformed when it propagates 8.16 km from the PL dam to cross-section S3, with a peak discharge of almost 3,000 $m^3$ $s^{-1}$ attained on 10 February at 9:00 and maintained for one day, until 11 February at 9:00. The peak discharge of the hydrograph generated on the D-PLD subbasin is slightly lower (around 2,550 $m^3$ $s^{-1}$) and occurs about 20 hours earlier (on 9 February 2023 at 13:00). Therefore, the total contribution of the D-PLD subbasin to the peak discharge arriving at Boane (cross-section S2) is relatively small, increasing the peak discharge roughly from 3,000 $m^3$ $s^{-1}$ to 3,500 $m^3$ $s^{-1}$. On the other hand, it generates a second peak in the hydrograph at the end of the event, increasing the maximum discharge on 13 February 2023 from 750 $m^3$ $s^{-1}$ to 2,400 $m^3$ $s^{-1}$ (Figure 10). This second peak of discharge is due to a second precipitation peak on 12 February 2023 (Figure 9), which is controlled by the reservoir in the U-PLD basin, but not in the unregulated D-PLD basin.

There are no field estimations of the discharge arriving at the Boane district during the flood event, and hence the previous hydrographs cannot be compared to observations. Only the maximum water depths estimated during the post-event field campaign at 20 locations in the AOI (Sect. 3.2.2) have been used to assess the numerical predictions (Figure 11). The comparison between the observed and predicted water depths has a Mean Error (ME) of 0.50 m and a Mean Absolute Error (MAE) of 1.06 m. The MAE is of the same order of magnitude as the vertical accuracy of the Copernicus DEM, which was estimated to have a global Root Mean Square Error (RMSE) of 1.7 m for terrain slopes lower than 20% (AIRBUS, 2020). Moreover, the positive ME means that the numerical predictions of the maximum water surface elevation have a positive bias with regard to the field estimations. Several reasons might explain the positive bias. First, within the main river channels the estimated terrain elevation given by the DEM may be higher than real, due to its limited spatial resolution and the fact that satellite-derived DEMs do not capture the terrain elevation below the water level. This issue cannot be solved in the absence of a huge amount of field topographic data, which is not available in data scarce regions. Second, the fact of having chosen a CN associated to wet AMC might have led to an overestimation of its value and in turn, to a subestimation of the infiltration rate. As mentioned in section 3.1.4, we considered wet AMC conditions because the 5-day antecedent rainfall depth was greater than 50 mm. Third, the water marks identified in the field work might underestimate the real maximum level reached by the water, since the fact that there is a mark at a certain location means that the water reached that level, but the flood might have reached a higher level without leaving a significant mark. These three factors might have contributed, to different degrees, to the deviations shown in Figure 11, the first two being probably the most relevant. Despite these limitations, it can be concluded that the numerical predictions of maximum water depths during the event in the AOI follow the same trend as the observations (Figure 11).

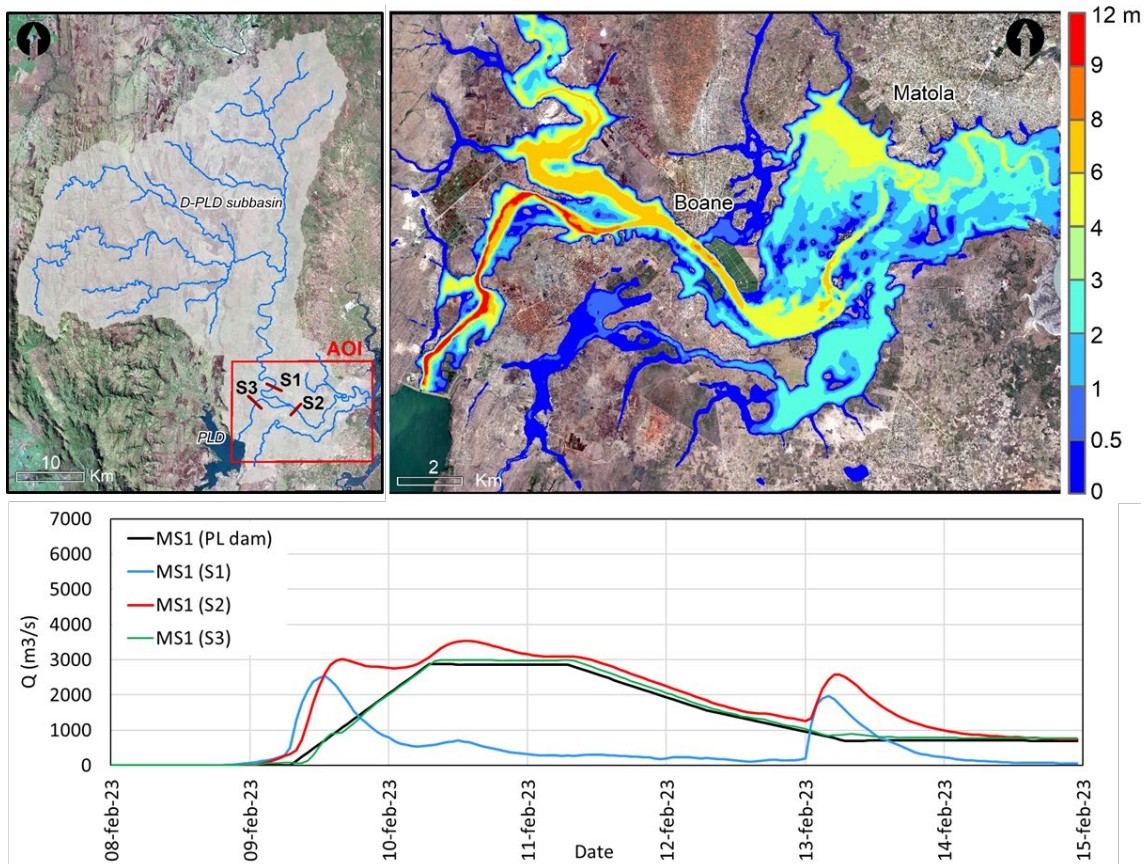

Figure 10: Extension of the D-PLD subbasin with its main river network and the location of cross section S1-S3 (top-left), map of maximum water depth in the AOI (top-right) and hydrographs computed with Iber at several cross-sections located around the confluence of the rivers Umbeluzi and Movene for the scenario MS1. Background images © Google Earth V 7.3.6.9345, http://www.earth.google.com [13 April 2023].

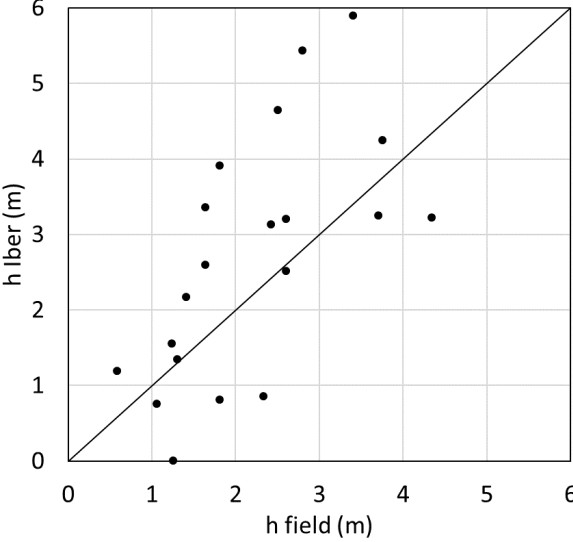

Figure 11: Observed vs. computed maximum water depths at the locations indicated in Figure 8. The identity line (black dashed) is shown.

### 4.1.3 Flood extent

The only available satellite image during the flood event was taken on 14 February at 3:00. At that time the peak discharge had already passed, and the flood was receding. The discharge at cross-section S2, estimated from the numerical model, was 915 m³ s⁻¹, almost 4 times lower than the maximum discharge reached at this cross-section during the whole event. Nonetheless, several floodplains were still covered by water in the AOI, as shown in Figure 12, which shows the water extent at that time,

as estimated using Iber (assuming as flooded areas those with a water depth greater than 0.1 m) and from the Sentinel-1 image, as well as the overlapping between these.

Most of the observed flooded areas are correctly predicted by the model, which shows a *HR* of 0.96. On the other hand, the model predicts several flooded areas that are not identified from the analysis of the Sentinel-1 image, the *FAR* being 0.37 and the *CSI* being 0.67. These results indicate that the model tends to overestimate the extent of the flood. A good portion of the overestimation occurs in the floodplains of the Movene tributary, upstream its confluence with the Umbeluzi. At the control point located in this region (ID 6 in Table 3 and Table 5) the model largely overpredicts the water depth (5.4 m versus 2.8 m). This reach comes directly from the northern D-PLD subbasin and thus, the flood extent in its floodplains is barely affected by the discharge of the dam, so the overestimation of the water depth and flood extent is probably an effect of an underestimation of the infiltration rate in the model.

The fact that the riparian vegetation can mask the water surface in floodplains with small water depths can also contribute to the difference between the modelled and satellite-derived maps, increasing the number of False Positives and thus the *FAR*. This limitation of satellite-derived flood maps might be, in certain cases, alleviated by the use of exclusion maps that identify the regions in which the satellite-derived estimation is not reliable (Zhao et al., 2021; Di Mauro et al., 2021).

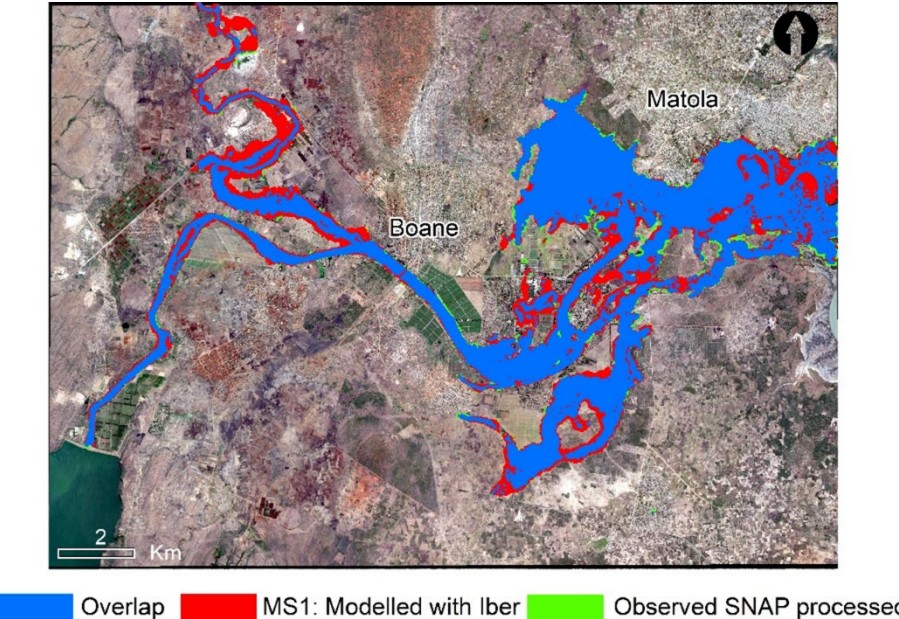

**Figure 12: Overlapping (blue) between the modelled (red) and satellite-derived (green) flood extent in the AOI on 14 February 2023 at 03:20. Background image © Google Earth V 7.3.6.9345, http://www.earth.google.com [13 April 2023].**

**4.2 Mitigation of the flood hazard by the PL reservoir. Scenario MS2**

During the flood event, from 8 February to 15 February, the PL reservoir received over 800 hm$^3$ from its upstream basin. Around 40% of this volume (330 hm$^3$, which is almost the maximum storage capacity of the whole reservoir) reached the reservoir within 24 hours on 10 February 2023. Thus, the PL dam had to release water to its maximum capacity (circa 3,000 m$^3$ s$^{-1}$) for reasons of structural safety and storage capacity for 2 days (10-11 February 2023). This high discharge, maintained over the course of 2 days, had an effect on the flood extent in the district of Boane, leading to strong criticism from the local communities and the press of the reservoir's management. However, due to the high inflow discharges relative to the reservoir storage capacity, there was no margin to diminish the outflow peak discharge.

Nevertheless, the reservoir had a net positive effect on the impact of the flood, reducing the extent of the affected areas, as shown in Figure 13, which compares the water extent computed under the real management of the reservoir during the event (scenario MS1) with the water extent that would have taken place if the PL reservoir had not existed (scenario MS2). In the absence of the reservoir (i.e. no flood control at all), the peak discharge of the hydrograph arriving at the district of Boane (cross-section S2) would have reached 6,000 m$^3$ s$^{-1}$, which is 70% higher than the discharge computed under scenario MS1

(3,500 m³ s⁻¹). We might note that the volume of the hydrographs arriving at Boane under scenarios MS1 and MS2 is roughly the same (Figure 13), and thus the effect of the dam was to redistribute over time the total volume of water arriving at the reservoir.

In terms of maximum flood extent and water depth during the event, the effect of the reservoir was to reduce the flooded surface in the AOI from 94 km² to 84 km², while the average water depth in the AOI was reduced from 2.9 m to 2.1 m (Table 5).

In order to relate the results of scenario MS2 to the maximum discharges arriving at Boane if the PL dam had not existed, we retrieved the maximum annual discharges at cross-section S2 from 1955 to 1986 from Lacamurima (2003). In that cross-

455 section there was a stream gauge, known as Boane hydrometric station E-8, which was in operation until 1986 when the PL dam was built. The maximum discharge registered in the E-8 Boane station was 7,250 m³ s⁻¹ on 30 January 1984. It is interesting to note that this value is in the order of magnitud of the peak discharge computed for the February 2023 flood under scenario MS2 (i.e. in the absence of the PL reservoir).

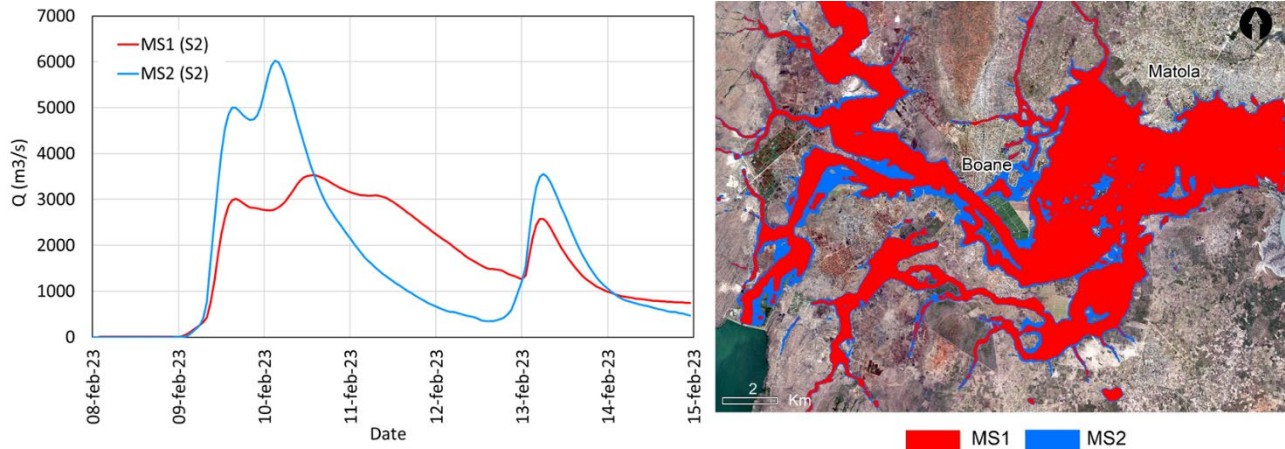

**Figure 13: Hydrograph arriving at Boane (left) and flood extent in the AOI, under the scenarios MS1 (actual management of the reservoir during the flood event) and MS2 (absence of PL reservoir). Background image © Google Earth V 7.3.6.9345, http://www.earth.google.com [13 April 2023].**

| ID | hmax (m) | | |
|---|---|---|---|
| | **MS1** | **MS2** | **MS3** |
| 1 | 0.0 | 0.6 | 0.0 |
| 2 | 3.2 | 5.0 | 0.0 |
| 3 | 3.2 | 5.3 | 0.0 |
| 4 | 4.6 | 6.7 | 0.4 |
| 5 | 2.5 | 3.6 | 1.6 |
| 6 | 5.4 | 6.6 | 5.3 |
| 7 | 4.3 | 5.7 | 3.2 |
| 8 | 6.0 | 7.6 | 5.0 |
| 9 | 2.6 | 3.8 | 1.5 |
| 10 | 0.8 | 2.3 | 0.4 |
| 11 | 3.4 | 4.9 | 2.6 |
| 12 | 3.9 | 6.5 | 0.2 |
| 13 | 3.2 | 5.3 | 0.0 |
| 14 | 0.9 | 1.6 | 0.5 |
| 15 | 2.2 | 2.6 | 1.9 |
| 16 | 0.8 | 0.9 | 0.5 |
| 17 | 1.4 | 1.6 | 0.9 |
| 18 | 1.2 | 1.6 | 1.0 |
| 19 | 1.6 | 2.3 | 1.2 |

| | 20 | 3.1 | 3.9 | 2.5 |
|---|---|---|---|---|
| | **Mean** | **2.7** | **3.9** | **1.4** |

**Table 5: Water depth at the 20 control points, computed with the numerical model, in the scenarios MS1, MS2 and MS3.**

### 4.3 Flooding from the unregulated subbasin. Scenario MS3

Figure 14 shows the water depth maps computed in the AOI under the scenarios MS1 (actual flood) and MS3 (contribution only from the unregulated D-PLD subbasin). If the dam were not to have spilled any water at all during the flood event, the peak discharge in the district of Boane would have diminished from roughly 3,500 m³ s⁻¹ to 2,300 m³ s⁻¹, the flooded area within the AOI would have been 76 km² instead of 84 km², and the average water depth in the AOI would have been 1.6 m instead of 2.1 m (Table 6). Despite this virtual reduction in the flood hazard, the damage to the population would have remained

very severe, since many populated areas in the region would still have been flooded to significant water depths, as shown in Figure 14. In any case, it should be stressed that, regardless of how the dam was managed, it would never have been possible to achieve scenario MS3 for the flood of February 2023, due to the limited storage capacity of the reservoir.

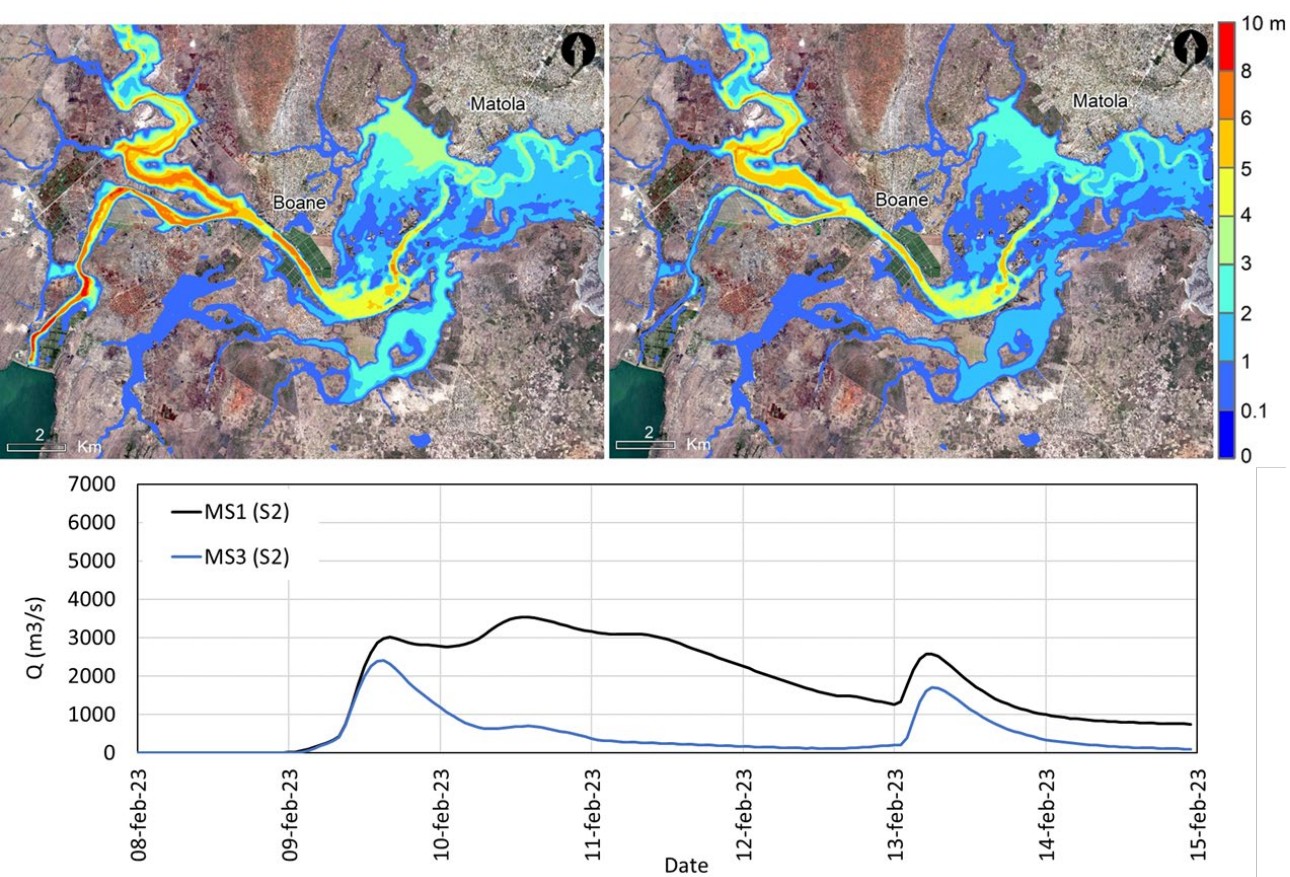

**Figure 14: Water depth maps computed under the scenarios MS1 (left) and MS3 (right). The hydrographs computed in cross-section S2 for both scenarios are also shown. Background images © Google Earth V 7.3.6.9345, http://www.earth.google.com [13 April 2023].**

| Scenario | Qmax in S2 (m³ s⁻¹) | Flooded area in AOI (km²) | Average depth in AOI (m) |
|---|---|---|---|
| MS1 | 3,500 | 84 | 2.1 |
| MS2 | 6,000 | 94 | 2.9 |
| MS3 | 2,300 | 76 | 1.6 |

**Table 6: Results of maximum discharge in cross-section S2, flooded area and average depth in the AOI for the three modelling scenarios.**

**5 Conclusions**

The numerical simulation of the flood that took place on 11-14 February 2023 in the Umbeluzi river basin confirms that integrated hydrological-hydraulic models based on the 2D-SWE combined with global data sources are efficient tools in reproducing the flood hazard during extreme rainfall events in data-scarce catchments of several thousands of km$^2$. The model used here (a GPU-enhanced solver for the 2D-SWE including rainfall and infiltration processes) was able to reproduce reasonably the peak discharge and flood extent during the event, using only satellite-derived products of rainfall, topography, land use, and curve number as input data, all of these available on a global scale. The maximum water depths estimated in a field survey after the flood event were not so accurately reproduced by the model. This can be attributed to a number of factors, among which the uncertainty of the currently available global DEMs, the uncertainty on the numerical parametrization of the infiltration losses and the uncertainty of the water depths estimated on the post-event field survey.

The methodology followed in this work is reproducible anywhere, but not necessarily with the same rate of success in all cases. It is expected to perform better in extreme rainfall events over wet terrains, occurring in relatively large catchments with a low rate of anthropization, as it was the case in the event analyzed in this work.

The spatial rainfall pattern over the basin shows that the highest cumulative rainfall depth during the event occurred around the PL reservoir, near the outlet of the U-PLD subbasin, contributing to reducing the response time of the basin and increasing the peak discharge into the PL dam. The quick response of the basin did not give any possibility of operating the dam spillways by releasing water with the aim of providing more storage in the reservoir available to mitigate the peak flow. Nevertheless, considering the basin size, a different rainfall distribution may provide this operational time if a hydrologic/hydraulic operational tool is implemented to forecast river discharges in real time, which would constitue a best practice for reservoir flood management.

In addition to the management of the PL dam during the flood event, two additional scenarios were modelled: in the case of the reservoir not existing (i.e. no flood control by the dam) and where the reservoir would control all the inflow hydrograph. From the results of these scenarios, it can be concluded that the PL reservoir contributed to reduce the flood hazard in Boane during the February 2023 event, reducing the peak discharge from 6,000 to 3,500 m$^3$ s$^{-1}$, the flooded area from 94 km$^2$ to 84 km$^2$, and the average depth from 2.9 m to 2.1 m. Even if there had been a reservoir capable of absorbing the entire volume of the hydrograph generated in its upstream basin, the extension of the inundation in the AOI would have been 76 km$^2$, with an average depth of 1.6 m.

The results presented here show that it is currently possible to delineate flood hazard maps in data-scarce regions under different scenarios using freely available numerical tools and input data, thus contributing to more efficient flood risk management. However, the accuracy of the water depth results might be limited by the spatial resolution and accuracy of the global DEMs currently available. Future enhancements in these global topography products will further improve the accuracy of the modelling approach presented here.

**Data availability**

The DEM of the Umbeluzi catchment is publicly available via the Copernicus Space Component Data Access PANDA Catalogue (https://panda.copernicus.eu/web/cds-catalogue). The rainfall data is publicly available via the Goddard Earth Sciences Data and Information Services Center (GES DISC) from NASA (https://disc.gsfc.nasa.gov). The ESA WorldCover 10m 2021 v200 map is publicly available via the ESA GlobCover Portal (http://due.esrin.esa.int/page_globcover.php). The Curve Number map is publicly available from the open repository figshare (https://doi.org/10.6084/m9.figshare.7756202.v1). The satellite image from Sentinel-1A is publicly available via the Copernicus Open Access Hub (https://scihub.copernicus.eu/dhus). The numerical simulations were done with the software Iber v3.1, which can be downloaded for free of charge from www.iberaula.com.

## Author contributions

MA built the numerical model and performed all the numerical simulations. LC supervised and revised the numerical model. LC and MA analysed the numerical results and their validation with the observed data. LC and MA wrote the first draft of the manuscript. JP revised the manuscript.

## Acknowledgements

The authors would like to thank the Head of the Water Resources Department of the Regional Water Administration of Southern Mozambique, Dr. Lizete Días, and their technicians Ernesto Tivane and Leonel Bila, for the data provided from the Pequenos Libombos Dam, as well as for the field work carried out in the Boane district in the days following the flood. The authors also acknowledge Victor Penas and the management staff from the AquaMoz project (Augas de Galicia/Xunta de Galicia, Cooperación Galega and Ingeniería sin Fronteras Galicia) for additional support provided.

## Competing interests

The authors declare that they have no conflict of interest.

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
