# Peer review of "Using integrated hydrological-hydraulic modelling and global data sources to analyse the February 2023 floods in the Umbeluzi catchment (Mozambique)"

_EGUsphere, 2023_

## Author Comment (AC1)

The manuscript presents a numerical model for a recent flood event in Mozambique; furthermore, two counterfactual scenarios show what would have happened if a dam that was used to mitigate the flood (Pequenos Libombos dam) had been either larger or absent. While the hydrological/hydraulics model is a standard one, the efficiency of the GPU-based implementation is striking.

However, I am doubtful about the merit of the material as a scientific research paper. The introduction gives the impression that this will be a report of the flood event. As a matter of fact, even though a section entitled "case-study" will follow, most of the introduction is focused on the flood and the role of the dam. At the end of the introduction, the key statements are that (1) hydrodynamic modelling is useful for management purposes and (2) the effect of the flood would have been less if more water had been stored and vice versa, both sounding quite trivial. In order to engage a reader, the manuscript would need more. Which are the challenges of performing a study like this? Which are the distinctive features of this work compared to others? What can be transferred to practitioners and stakeholders? I was hoping that the rest of the manuscript would provide this, but it is actually limited to the report of the results of the simulations for the real scenario and the two additional ones. The only statement that gives a bit more is that a study performed using just open data can be enough reliable, and this needs to be emphasized even though it is not completely new. In summary I strongly encourage the authors to try to enhance the scientific merit of their study.

We disagree with the overall assessment of Dr. Radice regarding the interest and novelty of our work. The two key statements mentioned by Dr. Radice are certainly quite trivial, but those are not the main conclusions stated in the manuscript.

First, we are not using a standard hydrological model (as it can be a lumped or semidistributed model as HEC-HMS, or a 2D distributed hydrological model based on the kinematic or diffusive wave equations), but rather an integrated hydrologic-hydraulic modelling approach based on the 2D shallow water equations (2D-SWE) with an enhanced GPU implementation. The model computes rainfall-runoff transformation in the hillslopes and river flow simultaneously in the whole catchment (discharges, water depths and water velocities). Even if this modeling approach is starting to be used in some studies, it is relatively recent and cannot be considered as standard practice. Moreover, we are not aware of any publication in which this modelling approach is applied to a real extreme flood event in a 5,000 km2 regulated catchment, using only satellite-derived data (i.e. no calibration of empirical parameters with in-situ data), and validated with 3 different kind of data: 1) time series of observed discharges; 2) water marks left by the flood; 3) satellite imagery of the flood extent.

Second, the conclusion of the work is not simply that "the effect of the flood would have been less if more water had been stored and vice versa", as Dr. Radice states. We provide a detailed quantification of the flood control exerted by the reservoir during a flood event that is one of the largest in the last 40 years (as we mention in section 4.2, the peak discharge estimated for this event was 6,000 m3/s, while the maximum discharge registered in the Boane station was 7,250 m3/s on 30 January 1984). In the "Conclusions" and "Introduction" sections we don't give the specific numbers on the effect of the reservoir, which are given in section 4.

Third, we consider that the case study analysed in the manuscript is in itself interesting due to its magnitude (as justified above) and that it occurred very recently. Our analysis can give some additional insight on the flood hazard in an extremely vulnerable region. This is why we refer to the consequences of this event and the general exposition to flood damage in Mozambique in the introduction. Both, the methodology and the analysis of this specific event are, in our view, interesting for the community.

Regarding the specific questions raised by Dr. Radice:

**Which are the distinctive features of this work compared to others?** As already mentioned above and in the manuscript, the distinctive features are:

1) the integrated hydrological-hydraulic model used, based on a GPU-enhanced implementation of the 2D-SWE.

2) the modelling approach using only satellite-derived global and freely available data and software, with no specific calibration (an approach that could be used anywhere with no requirements of local data)

3) the validation of such a modelling approach using 3 kind of data: discharge time series, water depth marks and satellite imagery

**Which are the challenges of performing a study like this?** We consider that the methodology followed in our work and its detail assessment (as mentioned in the answer to the previous question), is in itself a challenge that can be of interest to the scientific community

**What can be transferred to practitioners and stakeholders?** A methodology to quantify flood hazard in data scarce regions using only freely available data and software.

Apart from the general issue, the manuscript is generally well written and easy-to-read, thus I have just few detailed comments (below).

184: the CN value is quite high, probably due to the consideration of a wet soil in the AMC. Was this the case (soil already wet) for this event, based on available records? I mean, apart from the fact that it will later give a good performance of the model.

Yes, as mentioned in section 3.1.4, we considered CN values corresponding to wet AMC conditions because the 5-day antecedent average rainfall depth in the basin was greater than 50 mm, which is the threshold commonly used to distinguish between normal and wet AMC conditions when applying the SCS-CN method.

212: this width of the river section would indicate that the used DEM is detailed enough to have a few points within the river (at least for the high-order stems), which is good. However, it seems that no correction was applied to the DEM, therefore the terrain elevation may be higher than real. It can be mentioned that this issue simply cannot be solved in the absence of a huge data availability.

We agree. Since the aim of the study was to assess what can be modelled using only global and freely available data, we did not modify the DEM in the streams. Moreover, we don't have any local data in order to perform such DEM corrections with a minimum representativeness of the real depth of the streams. However, considering the magnitude of the flood (most of the water during the peak of the event flows through the floodplains) the effect of lacking a detailed definition of the river bathymetry is probably low.

Actually, the fact that "the terrain elevation may be higher than real" is coherent with the positive Mean Error of 0.5 m obtained when comparing the observed and predicted water elevations at the 20 control points (section 4.1.2), i.e. the model tends to overestimate the water elevation, also for the reasons stated in the manuscript. This will be mentioned in the revised manuscript.

249: please specify if this water elevation was maintained constant or was changed based on available information during the event.

The water elevation at the outlet boundary was maintained constant during the simulation. This will be mentioned in the revised manuscript.

303: an equation is missing for the F_1 score (that, besides, does not seem to be used in the following as in the paragraph of line 385 only the HR and the FAR are mentioned).

This is a mistake in the text. We considered that the HR and FAR indices were enough to quantify and interpret the comparison between the observed and modelled water extent. We will remove the reference to F1 in the revised manuscript.

359: it sounds strange that the D-PLD contribution generates a second peak at the end of the event if (line 357) the hydrograph is "earlier" than the release from the dam. Should be mentioned that this, evident from Fig 10, must be due to a second precipitation peak in the lower basin.

Yes, this is the case. The second discharge peak in S2 is due to the second precipitation peak. The discharge generated by this precipitation peak in the U-PLD basin is controled by the reservoir, but this is not the case in the unregulated D-PLD basin. We will mention this in the revised manuscript.

370: while this may be true considering the extent of the model and the resolution applied, it could be acknowledged that in some cases we see overestimation by 2 to 2.5 m (around 100% of a value determined from the water marks).

Yes, this is true and it is already shown in Figure 11. However, due to the inherent uncertainty on the water marks estimations, and given that we compare water depths at 20 control points, we consider that the quantification of the agreement between model and observations should be based on statistics of the error, as it is done in lines 363-369. The fact that at some points the error is over 2 m is as relevant as the fact that at some locations the error is as low as 0.05 m. None of these extremes are relevant given the uncertainty on the depth estimations derived from the water marks. In our opinion, the MAE derived from 20 samples is much more relevant for the overall assessment of the results.

---

## Author Comment (AC3)

**General comments**

**RC2 GC1.** The paper provides the use of a hydrologic-hydralic modeling framework, exploiting a fully shallow water equation solutor (implemented in the IBER software) combined with a SCS-CN runoff model in an ungauged basin in Mozambique which was hit by a severe flood in February 2023. The application is performed by using freely available source of information, i.e. global datasets for DEM, Land cover, CN parameters and precipitation. The basin also includes a large artificial reservoir so that the extent of the flood is affected by spillway regulation whose behavior is analyzed. The paper is well written and organized and the topic is very relevant with both flood management in areas provided with scarce hydrologic/hydraulic monitoring and also with respect to reservoir management.

As a general comment, I believe that the paper results are interesting and satisfactory with respect to an application developed in a data-scarce environment, nevertheless I would suggest the authors putting maybe less emphasis on the goodness of results achieved and more on the uncertainties related to them, for the reasons explained in following points.

We thank the reviewer for his positive assessment of our work.

**RC2 GC2.** Also, the results of the study case are satisfactory, as just said, but not necessarily easy to straightfordly address any data-scarce basin in the world. The methodology is, as they say, "reproducible anywhere" but, I would say, not necessarily with the same rate of success. The success of the application is probably also due to the particular study case which is performed in a (I would say) "enough large" basin (almost 5,500 km^2), with a very low rate of anthropical effect (built area is less than 2%) and a homogeneous topography namely, "a very flat topography".

We agree with the reviewer in the fact that the methodology is reproducible anywhere, but not necessarily with the same rate of success. The agreement with reality will depend on the factors mentioned by the reviewer. The methodology will probably perform better in events with high rainfall rates and a wet antecedent soil moisture content. We have clarified this in the conclusions of the revised manuscript in order to avoid misleading the potencial readers.

**RC2 GC3.** Moreover, the model validation was possible due to the availability of the Sentinel 1 image of flood extent and by ground observations of maximum water levels in a number of points for the flooded area. To my personal experience, both of these datasets are not always easy to find even in Europe or many other areas of the world. That said, the effort they have provided in finding global databases useful for the application is really noteworthy and also, I would agree that the kind of information they have used can be suggested for best practices of "after event" flood management in any area of the world.

It is true that the datasets used for validation are not easy to find, specially in African countries. But those are not needed to implement the proposed methodology. We have just taken advantage of their availability on this event in order to analyse and validate the proposed methodology.

**Major comments**

**RC2 #1.** The authors provide three scenarios, MS1 (actual management of the reservoir), MS2 (absence of reservoir), MS3 (what if the reservoir was able to retain the entire flow of its upstream basin). The three scenarios provide interesting insights. The comparison between MS1 and MS2 provides that the presence of the dam had a beneficial effect on the flood extent and depth. The third one also testifies that even if the

reservoir was larger the AOI would have been flooded by the second tributary (D-PLD subbasin), nevertheless I notice that in this third case the average water level (1.6 m) is significantly lower than in the MS2 (and actual scenario (2.1 m). I agree that the amount of damage could have been not so different, nevertheless a water level less than the average human height could make a significant difference when life is threatened and has to be saved. Hence, I think it could be of interest to know what would have been the flood extent and (average and maximum) depth if, at the initial condition of the event, the reservoir was empty or if it was at a different level below the NPL, see comment #2 below, in order to see if it could be beneficial or not influential at all (as it probably is, due to its small capacity with respect to flood volume).

We understand the reviewer's suggestion and in fact, we thought a lot about the most convenient scenarios to analyse. We finally decided to include MS2 and MS3 as two limit scenarios on the potential effect of a reservoir on the natural flood regime in Boane, MS2 representing no effect at all, and MS3 representing the maximum effect. We are aware that other intermediate scenarios are possible, and we thought a lot about how these could be defined. The problem is that almost all the reservoir's volume can be controlled by its hydraulic structures, since the dam has bottom outlets, and the crest of the controlled spillways is at an elevation of 24 m, for which the reservoir's volume is of just 8 hm3. Therefore, considering an empty reservoir or an elevation equal to the crest of the spillways is virtually the same for the analysis of this flood event.

So it is complicated to establish a level of the reservoir that could represent a plausible initial condition. Even if such a level could be established (considering for instance an empty reservoir, as suggested by the reviewer), the outlet hydrograph would depend on the way in which the spillway's gates were operated. Moreover, we don't have detailed information about the hydraulic characteristics of the controlled spillways. Under this situation the amount of potential scenarios is huge, considering the different combinations of initial volume in the reservoir and outlet hydrographs.

That's the reason why we decided to keep only the three scenarios included in the manuscript. In our opinion adding new intermediate scenarios wouldn't add too much to the discussion about the role of the dam, and at the same time it would lengthen the content of the paper.

**RC2 #2.** Figure 7 suggests the need for more information about the operational and the structure of the dam spillway system. In facts, from the figure it appears that in the first four days of observation (6 to 9 February), for water levels up to almost 46.5 m, the daily outflow Qout is zero while, on the 12 and 15 of February for water levels well below 46.5, there is a daily outflow above 500 m^3/s. This observation suggests that the spillways are regulated by some movable device or other hydraulic system that probably were operated (manually or automatically) during the flood event. This is not clearly stated but I believe is necessary for a discussion about reservoir management. The paper does not provide detailed information about the structural and hydraulic operational system for water release. Also there is not a definition of the "Normal Pool Level" (NPL). In particular, it would be of interest to know if there is a minimum level for water release (below NPL) which could be operated by means of such a movable device and, if yes, what is the reservoir volume at that level. These elements could be useful to define a fourth scenario as I have suggested at comment #1

As the reviewer has correctly inferred from Figure 7, the discharge flowing through the spillways and outlets of the PL dam are controlled by movable gates (as we have already mentioned in our answer to comment **RC2 #1**). We have clarified this in the revised manuscript.

Unfortunately, we don't have any detailed information about the hydraulic behaviour of the controled spillways, but even if we had it we would need to know how they were operated in order to derive the outlet discharge as a function of the reservoir's level during the flood event. In fact, for our study the most relevant data are the discharges that were spilled by the dam during the event of February 2023. Those discharges were established by the regional water administration (ARA-Sul) according to the operational rules of the reservoir.

The only information about the discharge structures is that the crest of the spillways is at an elevation of 24 m (the reservoir's volume for that level is 8 hm3), and that bottom outlets can completely empty the reservoir for practical purposes. But we cannot relate the outlet discharge with the reservoir's elevation.

In summary, the data that was available to us for this study were the discharges that were actually spilled by the dam during the event of February 2023, which are those plotted in Figure 7.

The NPL is the maximum operation level, which corresponds to the maximum water level that can be attained during normal operation conditions (i.e. when there is no flooding).

**RC2 #3.** In section 3.3.3, line 304, a F1 score is mentioned as a combination of HR and FAR, but it is not further defined neither it is used throughout the paper. Also False Negative (FN) cells are mentioned but, if I am not wrong, there is not focus on them in the result sections. I may suggest the use of other indices such as the Critical Succes Index (CSI) and more. Maybe the authors could refine this section by extending the use of these metrics to other indices or explaining the reason why they only focused on HR and FAR.

The mention to F1 was a mistake, and it was removed from the revised manuscript. FN are mentioned because they are used to compute the HR in Equation (4).

Following the reviewer's suggestion we have included the CSI in the revised manuscript. We believe that these three complementary indices, together with Figure 12, are enough to discuss the agreement between the flood extent predicted by the model and derived from Sentinel-1.

**RC2 #4.** Figure 9 (right) I would add, besides (or replacing) the regression line, the 1-1 line. The regression line, in facts, provide a satisfactory index of determination but suggests a systematic underestimation of the hydrologic/hydraulic model with respect to observation. By this light I don't think the regression line provides a correct information. I see that all points but one are almost perfectly centered. Only in one day the daily average discharge is missed (11 February). This could be a lack of the measured precipitation which is almost absent on that day. To my knowledge CHIRPS values of precipitation may have a high rate of uncertainty and also the CN hydrological model used for evaluating infiltration is rather than perfect.

We agree with the reviewer's comment. In this case the 1-1 line is more relevant than the regression line. We have therefore replaced the regression line by the 1-1 line in Figure 9.

**RC2 #5.** Section 4.1.1. At line 367-369 it is stated that "the positive ME means that the numerical predictions of the maximum water surface elevation have a positive bias with regard to the field estimations, which is coherent with the fact that the water marks identified in the field work represent a minimum threshold reached by the flood". But at line 280 it is stated "At each point identified, the maximum water depth reached during the flood was estimated". The authors should clarify whether the points were related to minimum or maximum levels of water. I believe they are maximum levels as it would not make sense to perform a field map of minimum water levels reached by water. I would suggest that the positive bias may be due to a number of different explanations not excluded the hydrological model used for runoff generation. It is well known that the CN method may provide overestimation of both volume and rate of runoff. The upper left portion of Figure 10 provides a shaded area of runoff which is practically all over the D-PLD sub-basin, independently of rainfall intensity which looks quite low in some areas of the sub-basin. Even the rate of infiltration in Figure 9 (left) looks low, even considering the possible underestimation of rainfall which CHIRPS may provide as already said in comment #4. On the other hand, the high value of the vertical accuracy of the Copernicus DEM (RMSE= 1,7 m) is not good news, considering it is of the same order of magnitude of the average water depth (2.1 m in MS1, 2.9 m in MS2 and 1.6 m in MS3 as from table 6).

We acknowledge the reviewer for this interesting discussion. Regarding the first part of his comment, we have realized that the statements included at lines 280 and 367-369 might be misleading for the reader. The water depths given in the manuscript are certainly related to the maximum levels of water, since they were estimated from the water marks left by the flood. What we wanted to state in lines 367-369 is that these flood marks might underestimate the real maximum level reached by the water, since the fact that there is a mark means that the water reached that level, but it might have reached a slightly higher level without leaving a significant mark. We have tried to better express this idea in the revised manuscript.

Nevertheless, there might be other alternative explanations to the positive ME. In particular, as the reviewer suggests, in any hydrological model there is always a significant uncertainty related to the estimation of the infiltration parameters (in our case the CN). This is especially true if the results were obtained without any model calibration, as in this study. The associated error might be either positive or negative, but it might well be the case that in our simulations the error is positive considering that we have chosen to use a CN associated to wet AMC, since the 5-day antecedent rainfall depth in the basin was slightly greater than 50 mm (as stated in line 183). Considering the reviewer's comment, we have included this potential explanation in the revised manuscript.

The infiltration rate in Figure 9 is certainly low, partially due to the assumption of a wet AMC. Probably it was slightly higher in the real event. However, as stated in the manuscript, the aim of the study was to analyse the kind of predictions that can be made in data scarce regions in which there is no possibility of calibration and thus, the models must be used with the default parameters, as it was the case here.

We should also clarify that the shaded area in the upper left portin of Figure 10 does not represent the surface runoff, just the extension of the D-PLD subbasin. So it is not related to the rainfall intensity. This might have misled the reviewer, so we have clarified it in the revised manuscript.

We also agree in the fact that the vertical accuracy of the Copernicus DEM is rather high compared to the typical water depths during a flood event. But this is probably the most accurate global DEM that is currently freely available. In any case, it might also explain the differences between observed and modelled water depths, as already mentioned in the manuscript.

**RC2 #6.** Figure 10 (bottom). Here is probably my major concern. It shows the hydrographs computed with Iber at different locations but it seems that the MS1(S3) line is not an output of Iber but rather a linear interpolation of average daily discharges obtained by water levels registered in the reservoir (they are consistent with those shown in Figure 7 red line). As a result, the MS1(S2) line here is the sum of an hourly discharge plus a daily discharge interpolated over different values which appears to me as a critical point of the paper. If my considerations are correct I think this point needs re-evaluation by the authors. If we go back to Figure 9 (left) we see that the daily discharge value (3,780 m^3/s) flowing into the reservoir obtained from the IBER output subtends a much larger hourly peak (5,700 m^3/s). That is the same point (the daily average) we find in Figure 7 as the maximum value obtained by IBER as Qin (in light blue). As I noted before we do not know anything about the spillway size and structure and about hydraulic regulation devices but even considering a very high efficiency of such a structure it is hard to believe that the ratio between maximum Qout and maximum Qin is equal to 2700/5700=0.47. In order to sum up the hourly hydrograph of the IBER output from sub-basin D-PLD with the hydrograph of the spillway discharge the hourly distribution of Qout is needed as well. It should be ideally obtained by knowing the geometry of the spillway structure, and of the lake, to route the hourly IBER output Qin of Figure 9 arising from sub-basin U-PLD into the reservoir and then into the spillway in order to obtain an hourly Qout. If such information is not available at least a peak coefficient could be applied to the daily average value shown in Figure 7, or a feasible ratio between hourly values of maximum Qin and maximum Qout should be searched for. Obviously, such consideration also affects results shown as from scenario MS1(S2) (e.g. Figures 13 and 14). It is likewise obvious that, should the authors state that the dam is provided

not with a standard spillway system but with a strongly regulated discharge control system, my concern will be solved and with it also the last sentence of the next comment.

We understand the detailed analysis made by the reviewer. However, as mentioned in our answer to **RC2 #2**, the spillways are controlled by movable gates. The hydraulic behaviour of the outlets and spillways, as well as their operation and percentage of opening during the flood is unknown to us.

The only data that is available is the total discharge (m3/s) spilled by the dam during the event, which is represented in Figure 10 bottom (black line). This was the discharge imposed at the dam location in the D-PLD model, and that's the reason why the green line in Figure 10 bottom MS1(S3) has such a pattern. To further clarify, the discharge imposed in the D-PLD model was the one provided to us by the ARA-Sul as the one spilled by the dam.

At the same time, the discharge imposed seems to be coherent if we consider that the inlet discharge to the reservoir computed with Iber is higher than the outlet discharge of the dam (2,800 m3/s) for approximately 20 hours, during which the volume difference between the inlet (computed from Iber) and outlet hydrographs would be around 90 hm3. The volume of the reservoir the days prior to the event was around 330 hm3, while the maximum volume during the event was around 450 hm3. Those numbers are coherent.

Also, if we look at Figure 12, there is an almost perfect match between the flood extent computed by Iber and the one derived from Sentinel-1 in the river reach between the dam and the confluence of Movene and Umbeluzi rivers. We should note that the discharge in this reach is mainly the one spilled by the dam (since the contribution of infiltration and rainfall might be neglected in that small area). If there was a relevant error in the discharge spilled by the dam, the agreement would probably be worse.

We understand the concerns of the reviewer, but another assumption about the discharge spilled by the dam with the available data would be purely hypothetical and not necessarily more precise than the values used here.

**RC2 #7.** Figure 11. In this Figure I see that the northernmost point, ID 6 in Table 3, if am not wrong is the one that provides the highest overestimation (second highest value) in water level h obtained by IBER vs h from field observation: 5.4 m (table 5) vs 2.8 m (table 3) of water depth over the ground level). If I read well Figure 8 this is also the only one placed on a reach affected only by the flood of sub-basin D-PDL. Consider now the significant FAR value (0.37) found in section 4.1.3 and look at Figure 12. I see that a good portion of the False Positive cells affecting FAR are in the same northern reach coming from sub-basin D-PDL. Considering that the discharge coming from the reservoir outflow may be affected by an underestimation error (see #6) both the high FAR value and the overestimation of water depth in point ID 6 may be an effect of the overestimation of runoff arising from the use of the CN infiltration model. Such overestimation may compensate the underestimation of the daily outflow hydrograph from the reservoir in the remaining points.

The reviewer is right in his considerations about the areas in which the highest number of False Positives occur. These are the floodplains of the Movene tributary, upstream its confluence with the Umbeluzi. It is also true that in this area the flood extent is barely affected by the discharge of the dam. Therefore, as the reviewer suggests, the overestimation of water depth (and flood extent) in this area is probably an effect of an underestimation of the infiltration rate in the model. This is very related to the reviewer's comment **RC2 #5**, so we refer also to our answer to that comment. We have introduced this consideration in the revised manuscript, in the discussion of Figure 12 and of the FAR value obtained.

On the other hand, we don't think that the daily outflow hydrograph from the reservoir is underestimated. As we mentioned in our answer to the previous comment (**RC2 #6**), the reservoir discharge was provided by the water administration ARA-Sul, and the good agreement between the flood extents estimated from Iber and from Sentinel-1 in the river reach located just downstream the dam suggests that the discharge spilled by the dam is correctly imposed in the model. Lastly, there are also several areas with False Positives after the

confluence, between Boane and Matola (Figure 12). This suggests that the overestimation of the discharge of the Movene tributary is not compensated by an underestimation of the reservoir's discharge (or at least this cannot be easily inferred from Figure 12).

**RC2 #8.** the rainfall event that generated the flood of February 2023 was particularly severe also by the light of its spatial distribution. In facts, the presence of the highest rainfall intensity in areas close to the reservoir generated a very quick response that did not gave any possibility of operating on the reservoir by releasing water at a discharge compatible with river conveyance with the aim of providing more storage in the reservoir available to mitigate the peak flow. Nevertheless, considering the basin size and travel time of water, a different rainfall distribution may provide this operational time. I would suggest mentioning this possibility, practicable by mean of this hydrologic/hydraulic operational tool, as a discussion item for best practice in reservoir flood management.

We fully agree with the reviewer's statement. In fact, we mentioned in lines 143-148 that the spatial distribution of rainfall during this event contributed to reducing the response time of the basin, thus increasing the peak discharges flowing into the PL dam. Nonetheless, the comment made by the reviewer is more precise and very appropriate, so we have included it in the Conclusions section.

**Minor comments**

**RC2 #9.** Line 85. The CHIRPS acronym is only used in this line and it is not explained. I suggest expanding the acronym and explain in section 3.1.2 the relationship with GPM-IMERG.

Following the reviewer's suggestion, we have expanded the CHIRPS (Climate Hazards Group InfraRed Precipitation with Station data) acronym in the revised manuscript. In fact, we have only used CHIRPS to characterise the catchment in terms of average annual precipitation, because has rainfall estimations since 1980. However, it was not used as input data in the model because it only provides rainfall estimates with a maximum temporal resolution of 1 day, which is not enough for relatively short and intense events. In section 3.1.2. we describe the GPM-IMERG data set in detail because it is the one used in our study. We don't describe in detail other rainfall products because there are many others well described in the literature, and that would lengthen unnecesarily the section.

Nevertheless, considering the reviewer's suggestion we have included a brief mention to the spatial and temporal resolution of other freely available rainfall products in relation with GPM-IMERG.

**RC2 #10.** Figure 8. What is the shaded area in the background?

The shaded area represents the maximum flood extent for the scenario MS1, in order to have a reference about where the points are located with respect to the flooded area. We have specified it in the caption of Figure 8 in the revised manuscript.

**RC2 #11.** Line 426. The reference, if I am not wrong, should be to figure 14.

The reviewer is right. We have modified the reference in the revised manuscript.

---

## Author Comment (AC4)

The hydrological-hydraulic study presented in this paper attempts to reproduce a pluvial flood across the Umbeluzi catchment in February 2023. The numerical simulations were conducted with the free software Iber+, well known by the primary author because he is one of the original developers. The basin has a drainage area of approximately 5000 km2 and a mean slope of about 10%, which led to a peak discharge close to 3000-5000 m3·s-1 for the studied floods. The study's input data consists of the Copernicus GLO-30 Digital Elevation Model (DEM), 66 satellite-based GPM-IMERG rainfall database pixels, and the curve number (CN) data set GCN250. The spatial resolutions are, respectively, 30 m, 9 km and 250 m. For the validation step, the authors limited the analysis to the outlet region of the catchment using: i) a 10 m resolution Sentinel-1 image, taken on 14 February 2023 at 03:20 UTC, with a discharge of 915 m3s-1; ii) twenty (post-event) watermarks measured in field works on 20-21 March 2023.

**RC3 #1.** The paper is well presented, the materials and methods are explained briefly, referencing other literature for details, and the results are described concisely. I have no concerns regarding the writing and presentation. However, the size of the basin and the magnitude of the flood could be more exceptional regarding other studies also conducted with Iber+ by other authors not cited in the current version of the paper. The pluvial inundation in the Umbeluzi basin has no particular value because the peak discharge is not high for the catchment size; however, if the authors could show an essential novelty regarding the methodology from a broader scientific perspective, it would deserve publication.

First, the size of a catchment is not a criterion to assess the interest of a hydrological study. As stated in the introduction, the choice of this case study was not the size of the catchment, but rather the intense flood event that took place in the province of Maputo on February 2023, which resulted in severe damage to population, infrastructure and agricultural lands. These kind of catchments are quite frequent in Mozambique, a region with very limited availability of local hydrological data and limited access to computational resources. Thus, its is interesting to analyse to which extent a relatively recent integrated hydrologic-hydraulic modelling approach, based on the 2D shallow water equations, that can be run in a standard PC or laptop, combined with global and free databases, is able to reproduce extreme flood events under these conditions.

Regarding the magnitude of the event, as mentioned in the previous paragraph and in the introduction of the manuscript, it was one of the most extreme in the last 20 years, causing important economic losses and significant damage to infrastructure, agriculture and population. So it can certainly be considered a high discharge. Moreover, as mentioned in lines 415-419, the maximum discharge estimated for this event was similar to the maximum discharge registered in the Boane hydrometric station from 1955 to 1986 (i.e in a period of 31 years).

In any case, the reviewer does not seem to be aware about the relation between peak discharges and catchment size, otherwise he wouldn't make the statement "the peak discharge is not high for the catchment size". The following figure represents the estimated peak discharge in U-PLD for the event analysed in the paper (green circle), together with the maximum discharge recorded at the Boane hydrometric station in 1984 (red circle) and the envelope curve from the Maximum Streamflow Discharge of the European Rivers (blue line) (Herschy, 2002). The same figure shows more than 500 extreme events recorded in 21 European countries according to Med-Hycos (Mediterranean Hydrological Cycle Observing System). As it can be seen, the discharge of the event analysed is almost overlapping with the European envelope and is higher than all those observed in Europe for the same catchment size, which proves its exceptional magnitude.

[Figure]

**Figure 1.** Maximum discharges observed vs. size of the catchment.

The same conclusion could be drawn by following the method of Regional Maximum Flood Peaks in Southern Africa proposed by Kovacks (1988), which is based on the equations defining the nomograms proposed by Francou and Rodier (1967). The regional coefficient K obtained for the flow of 5,763 m3/s is 4.9, a value in agreement with that estimated by Kovacks (1988) for the region in which the Umbeluzi basin is located.

**RC3 #2.** Furthermore, the Conclusions are not supported by the Results. As explained below, it is impossible to achieve the Conclusions established in the last section of the paper because of the absence of more accurate input data.

We would ask the reviewer to be more precise in his comments. If the reviewer could tell us which conclusion in the paper is impossible to achieve, we would be able to either refute his argument or take it into consideration.

**RC3 #3.** Introduction. The Authors should cite other software for distributed hydrological simulations based on the two-dimensional Saint-Venant equations and GPU acceleration. In particular, TRITON (Morales-Hernández et al. 2021), SERGHEI-SWE (Caviedes-Voullième et al. 2023) and LISFLOOD-FP (Sharifian et al. 2023). Also, the Authors need to establish the limitations of Iber+, which only allows using one GPU. In contrast, other alternatives allow multi-GPU, precisely, to achieve the required spatial resolution in accurate distributed-hydrological simulations.

It is true that we could cite other solvers that implement acceleration techniques. We didn't do it in the first version of the manuscript because the study is not focused on numerical aspects. But we have done it in the revised manuscript.

We have included the following references to other solvers of the 2D-SWE that implement parallelization techniques, either on GPU or CPU: Noh et al. (2018), Xia et al. (2019), Sanders and Schubert (2019), Morales-Hernández et al. (2021), Caviedes-Voullième et al. (2023), Sharifian et al. (2023).

In our opinion, the parallelization for 1 GPU is not a limitation for the purpose of this study. First, because the computational time needed to run the simulations using a standard PC was around 20 minutes (for a period of 9 days of real time), which is not limiting at all. Second, because the results won't change using several GPUs, the simulation will just run faster depending on the number of GPUs used. Third, because running a code in multiple GPUs requires the use of very expensive computational clusters that are not generally available, specially in regions as Mozambique, thus, precluding their application. Moreover, most shallow water codes, even in Europe, are run in personal computers with only one available GPU. We recognize that having a code that can be run in multiple GPUs is an advantage, but as we argued, this is not relevant for this study. On the other hand, Iber is a freely available and widely used software for flood hazard estimation that can be easily run in any personal PC without the need of any additional pre- or post-processing software, which is not the case of TRITON or SERGHEI-SWE.

In any case, it is not the purpose of this study to establish a comparison of the computational efficiency, availability or ease of use of different 2D-SWE solvers. We have used Iber because we are the developers of the software, but of course, the same kind of results could be obtained with other similar software. For this reason, we have removed the single reference that there was to Iber in the Conclusions section, we just refer to codes that solve the 2D shallow water equations.

**RC3 #4.** Introduction. The limitations of the numerical study concerning the use of global data source and the limited amount of data for the validation has to be explicitly explained in the Introduction. Please note that the spatial resolutions you used, i.e., 30 m for DEM, 9 km for rainfall and 250 m for CN, are too coarse for flood hazard mapping using the 2D Saint-Venant equations. In Spain and other European countries, we have made great efforts and spent huge amounts of money to acquire LiDAR data with the accuracy required for accurate flood risk mapping (Díez-Herrero et al. 2009; Sánchez and Lastra 2011; Olcina-Cantos and Díez-Herrero 2021). Both in terms of spatial resolution and elevation errors, among other essential factors. The global data source used by the authors cannot yield accurate flood maps. Otherwise, why are we making so many efforts to accurately implement the EU Directive 2007/60 on the estimation and management of flood risk?

The reviewer doesn't seem to be aware about the peculiarities of doing hydrology in data scarce regions, and that there are many regions in the world which do not have the data availability that we have in Europe. Of course European countries have done a great effort in the last years to obtain accurate data (particularly DEMs) in order to implement the EU Floods Directive with the highest possible level of detail, because they have the means to do that. But this level of detail is not possible in most African and Latin American countries, which, by the way, are much more vulnerable to floods than European countries, and also need to protect themselves against floods. So, you need to use the best data available in each case. If the reviewer is aware of more accurate data available at the global scale (or even in Mozambique), we would be grateful to know.

Indeed, our study serves as a case study of what can be done with 2D-SWE models in data scarce regions using globally available data. This is clearly stated in the manuscript as one objective of the study, and we believe that our results can be valuable for the hydrological community.

Regarding the reviewer's comment on the "limited amount of data for the validation", we should refer to the following comment by Reviewer #2, which seems to be much more aware about the peculiarities of doing hydrology in data scarce regions: "*the model validation was possible due to the availability of the Sentinel 1 image of flood extent and by ground observations of maximum water levels in a number of points for the flooded area. To my personal experience, both of these datasets are not always easy to find even in Europe or many other areas of the world.*"

**RC3 #5.** Introduction. Please cite other studies using Iber+ and other software for flood risk mapping using GPU and distributed numerical simulations in basins of similar size. For instance, Moral-Erencia et al. (2021) computed and validated flood maps using Iber+ in a catchment of about 2000 km2, with a mean slope as steep

as for Umbeluzi, using a computational mesh with 20 million cells and sub-metric spatial resolution in some river stretches. Also, note that the satellite-based IMERG rainfall data set (the same one used by the authors) underpredicted the accumulated precipitation by 50% in such a study.

We are aware of the interesting work of Moral-Erencia et al. (2021), that we have now cited in the revised manuscript. In fact, we already had 12 references in the original manuscript to other studies solving the 2D-SWE at the catchment scale (using Iber or other software). In addition to the new references of Sanders and Schubert (2019), Morales-Hernández et al. (2021), Caviedes-Voullième et al. (2023) and Sharifian et al. (2023), there are now 17 references related to applications of the 2D-SWE for flood hazard mapping.

However, we would like to note that the case study and objectives of Moral-Erencia et al. (2021) are very different from ours. Their work is in a Spanish watershed very rich in data (for instance, they use a LIDAR-derived DEM with a spatial resolution of 2m and a vertical accuracy of 0.2m, which is available for the whole of Spain), i.e. it is not a data scarce region in which it is necessary to resort to global satellite data sets.

Regarding the number of cells in the computational mesh and spatial resolution, we refer to our detailed answer to the next reviewer's comment (RC3 #6). In any case, in our opinion there is no need to go to such high resolutions if the input data (and specially the DEM) has a much coarser resolution, as it is the case in data scarce regions (and in the Umbeluzi). The accuracy of a numerical simulation does not depend only on the resolution of the numerical mesh, specially in hydrological simulations, where the uncertainty introduced by the input data and parameters (rainfall, topography, infiltration, land uses, etc) is in general more significant that the uncertainty introduced by the mesh resolution.

Regarding the accuracy of the GPM IMERG rainfall data, we would just like to notice here again that the data availability in countries as Mozambique is not the same as in Europe. Despite its limitations and low accuracy when compared with other local rainfall products based on in-situ raingauges and meteorological radar, the GPM IMERG data (and other satellite rainfall products) are routinely used in hydrological studies in data scarce regions all over the world. Moreover, there are many publications in which the accuracy of GPM-IMERG is analysed in different regions of the world, giving a much comprehensive evaluation of this product than the one given in Moral-Erencia et al. (2021), which is limited to a specific basin of Spain and to a single storm event and thus, it lacks of generality. We will just to mention some references. Saouabe et al. (2020) evaluated the accuracy of the near real-time product (IMERG) compared to observed rainfall and its suitability for hydrological modeling over a mountainous watershed in Morocco and concluded that the GPM-IMERG precipitation estimates can be used for flood modeling in semi-arid regions such as Morocco and provide a valuable alternative to ground-based precipitation measurements. In China the capacity of the GPM product to capture the temporal variations of extreme precipitation was analyzed by Jingyu Liu et al. (2020) within the Yuan River Basin, concluding that the GPM-derived product reasonably estimated the flood characteristics. In Spain Tapiador et al. (2019) highlight that the use of GPM contributes extraordinarily to improve the monitoring of extreme events in near-real time, concluding that the GPM-IMERG compares well with observations in general for the major 2019 September floods in Spain. Also a recent review by Gosset et al. (2023) about the role of satellite observations for monitoring pluvial and fluvial flood in Africa highlighted that major recent flood events in Africa have been well depicted by satellite observations, illustrating the feasibility of satellite monitoring for better surveillance of the food risk in this region. In summary, when in-situ rainfall data is not available, GPM IMERG can be used (an is commonly used) as the one of the best alternatives to characterise extreme rainfall.

**RC3 #6.** Section 3.2 Numerical model. "The size of the mesh elements ranged from 25 m in the main river reaches to 80 m in the hillslopes" (Line 213) and "Considering both models and the whole Umbeluzi catchment, the total modelled surface was 5461 km2, and the total number of elements was approximately 2.6 million (Lines 223-224)". The computational grid is too coarse, even coarser than the DEM. The grid size affects as much as the physical parameters in distributed-hydrological simulations, see Caviedes-Voullième et al. (2012). Subsequently, an additional numerical simulation using between 20 and 40 million cells is required. The grid

convergence study is a standard requisite in any CFD simulation (Blocken and Gualtieri, 2012). In my experience, considering that the model is already configured in Iber+, this task is not time-consuming. The authors only need to refine the mesh to achieve the maximum number of cells a single GPU allows.

It is true that the grid size (if too coarse) affects the model results, but I wouldn't say "as much as the physical parameters". Once a certain mesh resolution is attained, the model results barely vary and are therefore not sensitive to further mesh refinements. On the other hand, the physical parameters and other input data (specially infiltration parameters and rainfall) can have a much larger influence on the results.

The reviewer cites Caviedes-Voullième et al. (2012) to support his assessment about the relative effect of the mesh size compared to the effect of physical parameters. First, we should notice that the study mentioned was done in one single catchment of 2.8 km2, modelling one single rainfall event, and with a model that was not able to reproduced the observed hydrograph at the basin outlet (in fact most of the numerical results represent synthetic conditions with no infiltration and no bed roughness, and the only case in which infiltration is considered does not match the observed hydrograph at the basin outlet). So the conclusions that can be drawn from such a study are very limited.

Having said that, if one looks at Figure 7 of Caviedes-Voullième et al. (2012), the difference in the simulated outflow (red line) obtained with the meshes SS5 (size of 5 m, circa 110k elements) and SS20 (size of 20 m, circa 7k elements) is very small. The peak discharge varies roughly from 17.5 m3/s to 18.5 m3/s and the computed hydrograph is very similar in both cases. A similar conclusion can be drawn from Figure 8, except for the fact that their model exhibits some numerical instabilities in some cases.

So, the conclusion that can be drawn from Caviedes-Voullième et al. (2012) is that a mesh size of around 20 m might be enough to model their case study with the 2D-SWE. And in fact, this is in agreement with the results of a study that we made recently on the effect of the DTM and mesh size resolutions when solving the 2D-SWE in hydrological applications (García-Alén et al., 2022). In that study, seven real rainfall events in four different catchments (from 0.3 km2 to 3,750 km2) were correctly reproduced with the numerical model. In the medium size catchments (40 km2 and 200 km2) a mesh resolution of 25 m produced almost the same hydrographs as a mesh resolution of 10 m, while in the largest catchment (3,750 km2, a similar size as the one analysed in this paper), the Mean Absolute Error (MAE) and NSE was almost the same with meshes of 25 m and 65 m resolution. In all cases the NSE was of the order of 0.9, and the MAE normalised to peak flow was below 5%, values that can be considered very satisfactory when modelling single events in such large catchments.

In any case, **to further reassure the reviewer, we have refined the mesh size**, using 30 m (the same resolution as the DEM) in the whole catchment (hillslopes and rivers) and the results barely change (see Figures 2 and 3 below). The number of elements of the new meshes are 9.5 M (instead of 1.5 M) in the U-PLD basin, and 4.4 M (instead of 1.1 M) in the D-PLD basin. Thus, the whole domain is discretised with almost 14 M elements (instead of 2.6 M). On the other hand, the computational time increases 4 times in the D-PLD basin and 6 times in the U-PLD basin.

Clearly from Figures 2 and 3, the differences between both models are not significant for the purpose of this study, specially considering the uncertainty associated to the other input data and model validation data. And in any case, those differences wouldn't change any conclusion of the study.

[Figure]

**Figure 2**. Mesh convergence analysis. Hydrographs computed at the outlet of the U-PLD basin in MS1 using computational meshes of 1.5 M and 9.5 M elements (left); water depths computed in MS1 at the 20 control points located in the D-PLD basin using computational meshes of 1.1 M and 4.4 M elements.

[Figure]

**Figure 3**. Mesh convergence analysis. Maximum water depths computed in MS1 in the AOI, using computational meshes of 1.1 M and 4.4 M elements.

Finally, we consider that the reference Blocken and Gualtieri (2012) is not appropriate in this case, since it deals with very different kind of flows: "natural ventilation of the Amsterdam ArenA football stadium" and "transverse turbulent mixing in a shallow water flow", i.e. nothing to do with hydrological modelling. In our opinion there are no modelling steps that should be followed always in any numerical model. Depending on the equations being solved (1D-SWE, 2D-SWE, 3D-RANS, LES, DNS, etc.) and on the specific application (aeronautics, reservoirs, coastal engineering, river flow, hydrology, etc.) the approach is of course different.

Moreover, many studies in which the 2D-SWE are applied at the basin scale do not present a mesh convergence analysis, see for instance Xia and Liang (2018), Xia et al. (2019), Morales-Hernández et al. (2021), or Moral-Erencia et al. (2021), just to mention some of the references cited by the reviewer. Mesh convergence analysis are most commonly included in studies in which the main objective is to present new numerical schemes or software developments, or in which simplified geometries are modelled.

**RC3 #7.** Equations (2)-(3). Why did you neglect the Reynolds stresses even in the main river?

Iber includes several depth-averaged turbulence models, but it is well-known that they are not relevant at all in this kind of hydrological simulations (i.e. at this spatial scales). In these cases, the flow resistance is characterised mainly by the bed roughness. In fact, I am not aware of any publication in which a hydrological simulation at the basin scale is done including a turbulence model to compute the Reynolds stresses. Again, in hydrology the modeller should decide which processes should be incorporated (or not) in the model. Not always the more is the better. There is no sense in including irrelevant processes, because you just increase the parametrisation and complexity of your problem, without any real advantage.

**RC3 #8.** Equation (4). Please also evaluate the Critical Success Index (CSI) by Bates and Roo (2000) to compare your value with other studies. The CSI is more common than HR and FAR.

We have included the CSI in the revised manuscript. HR and FAR are also commonly used and they have a simple physical interpretation, so we have also maintained those indicators.

**RC3 #9.** Figure 11. "Observed vs. computed maximum water depths at the locations indicated in Figure 8". The maximum absolute error in the computed water depth values is extremely high concerning the field measurements. For instance: hiber=6 m for hfield= 3.5 m, or hiber = 4 m for hfield=1.9 m. Such errors are too severe for a flood study. It shows clearly that the global data source is inaccurate for detailed flood risk mapping, contrary to the author's statements in the Conclusion section.

We agree with the reviewer in that point. Probably we have been too enthusiastic in some of the statements made on the conclusions. We have rewritten them, and we have acknowledged that the model was able to correctly reproduce the flood hydrograph and flood extension, but not so trustable in the predicted water depths.

Having said this, the error is the predicted water depth values is related to the vertical accuracy of the Copernicus DEM (which is the best globally available).

**RC3 #10.** Figures 12-14 and their corresponding descriptions: Why did you limit the AOI to the basin outlet? The inundation area is too broad and probably covers the whole floodplain (from a geological perspective). Hence, it is easy to match the observed and simulated flood maps. Please include a map of the DEM slope in such an area to check. Conversely, the D-PLD headwater should be more sensitive and exciting for validation. Indeed, other studies, such as Moral-Erencia et al. (2022), verified the inundation maps in the catchment, not only in the outlet region.

The AOI was chosen considering the area in which the flood of February 2023 generated more damage, which is also the most exposed and vulnerable area of the basin. As mentioned in the Abstract and Introduction, it was in Boane and in the surroundings of Maputo where the economic, agriculture and human losses were concentrated, due to the settlements, farms and transport infrastructure located in the floodplains around the confluence of the rivers Umbeluzi and Movene. This AOI extends over 310 km2, which is a quite large area, and it can be thought of as an Area of Potential Significant Flood Risk (APSFR), even if APSFR haven't been identified officially in Mozambique. Thus, the rest of the catchment does not have the same interest regarding flood damage. In addition, the field campaign to estimate water depths was only undertaken in the AOI, precisely for the above reason.

Also, we should notice that the reviewer is wrong in his general comment about the areas in which it is more easy to match the observed and simulated flood extent. It is precisely in the upper reaches of a catchment where the river cross-section is more confined by the topography and thus, the water extent is less sensitive

to errors in the predicted water depth. On the other hand, in flat areas a small error in the estimation of the water depth will produce a large error in the horizontal extension of the flood.

**RC3 #11.** Conclusions. The limitations of Iber+ for flood risk mapping in basins of 5000 km2 should be clearly stated. Commenting both on the inaccuracy of global data source for DEM and precipitation and also because of the maximum RAM of a single GPU, which limits the total number of cells in the computational grid (and hence, the spatial resolution).

The accuracy of the global DEM is already stated in the manuscript, and its potential influence on the water depth results obtained with the model has been mentioned in the Conclusions of the revised manuscript. We cannot draw any conclusion about the accuracy of the GPM precipitation data because we don't have field data to compare with. At the same time there are too many publications in which the accuracy of GPM-IMERG is analysed in different regions of the world, so this information is already accessible for any reader. We refer to the last paragraph of our answer to **RC3 #5**.

Regarding the parallelisation on a single GPU and the number of computational cells, as mentioned in our answer to previous comments, we don't believe it is a limitation of the methodology. Having said that, the conclusions are not about a specific software, but about a modelling approach. Other software solving the same equations with the same input data could be used probably with similar results and conclusions. We have clarified this in the conclusions section, avoiding to mention Iber+.

**References**

Blocken, B.; Gualtieri, C. (2012). Ten iterative steps for model development and evaluation applied to Computational Fluid Dynamics for Environmental Fluid Mechanics. Environ. Model. Softw. 2012, 33, 1–22.

Caviedes-Voullième, D.; Morales-Hernández, M.; Norman, M.R.; Özgen-Xian, I. (2023). SERGHEI (SERGHEI-SWE) V1.0: A Performance-Portable High-Performance Parallel-Computing Shallow-Water Solver for Hydrology and Environmental Hydraulics. Geosci. Model Dev., 16, 977–1008.

Francisco J. Tapiador, Cecilia Marcos, Juan Manuel Sancho, Carlos Santos, José Ángel Núñez, Andrés Navarro, Chris Kummerow, Robert F. Adler. The September 2019 floods in Spain: An example of the utility of satellite data for the analysis of extreme hydrometeorological events. Atmospheric Research, Volume 257, 2021, 105588, ISSN 0169-8095, https://doi.org/10.1016/j.atmosres.2021.105588.

Francou, J., & Rodier, J. A. (1967). Essai de classification des crues maximales observées dans le monde. Cahiers de l'ORSTOM, série Hydrologie, 4(3): 19–46.

García-Alén, G., González-Cao, J., Fernández-Nóvoa, D., Gómez-Gesteira, M., Cea, L., & Puertas, J. (2022). Analysis of two sources of variability of basin outflow hydrographs computed with the 2D shallow water model Iber: Digital Terrain Model and unstructured mesh size. Journal of Hydrology, 612, 128182.

Gosset, M., Dibi-Anoh, P.A., Schumann, G. et al. Hydrometeorological Extreme Events in Africa: The Role of Satellite Observations for Monitoring Pluvial and Fluvial Flood Risk. Surv Geophys 44, 197–223 (2023). https://doi.org/10.1007/s10712-022-09749-6

Herschy, R. W. (2002). The world's maximum observed floods. Flow Measurement and instrumentation, 13(5-6), 231-235.

Jingyu Liu, Juan Du, Yumeng Yang & Yini Wang (2020) Evaluating extreme precipitation estimations based on the GPM IMERG products over the Yangtze River Basin, China, Geomatics, Natural Hazards and Risk, 11:1, 601-618, DOI: 10.1080/19475705.2020.1734103

Kovacs, Z. P. (1988). Regional maximum flood peaks in Southern Africa. Technical Report TR 137. Pretoria: Department of Water Affairs, Directorate of Hydrology.

Morales-Hernández, M.; Sharif, M.B.; Kalyanapu, A.; Ghafoor, S.K.; Dullo, T.; Gangrade, S.; Kao, S.; Norman, M.R.; Evans, K.J. TRITON: A Multi-GPU Open Source 2D Hydrodynamic Flood Model. Environ. Model. Softw. 2021, 141, 105034.

Noh, S. J., Lee, J. H., Lee, S., Kawaike, K., & Seo, D. J. (2018). Hyper-resolution 1D-2D urban flood modelling using LiDAR data and hybrid parallelization. Environmental Modelling & Software, 103, 131-145.

Sanders, B. F., & Schubert, J. E. (2019). PRIMo: Parallel raster inundation model. Advances in Water Resources, 126, 79-95.

Saouabe, T.; El Khalki, E.M.; Saidi, M.E.M.; Najmi, A.; Hadri, A.; Rachidi, S.; Jadoud, M.; Tramblay, Y. Evaluation of the GPM-IMERG Precipitation Product for Flood Modeling in a Semi-Arid Mountainous Basin in Morocco. Water 2020, 12, 2516. https://doi.org/10.3390/w12092516

Sharifian, M.K.; Kesserwani, G.; Chowdhury, A.A.; Neal, J.; Bates, P. LISFLOOD-FP 8.1: New GPU-Accelerated Solvers for Faster Fluvial/Pluvial Flood Simulations. Geosci. Model Dev. 2023, 16, 2391–2413.

Xia, X., Liang, Q., & Ming, X. (2019). A full-scale fluvial flood modelling framework based on a high-performance integrated hydrodynamic modelling system (HiPIMS). Advances in Water Resources, 132, 103392.

---

## Author Response (AR2)

**RESPONSE TO REVIEWERS COMMENTS ON THE MANUSCRIPT egusphere-2023-1003**

**Title:** Using integrated hydrological-hydraulic modelling and global data sources to analyse the February 2023 floods in the Umbeluzi catchment (Mozambique)

**Authors**: Luis Cea, Manuel Álvarez and Jerónimo Puertas

**GENERAL COMMENTS**

The authors would like to thank the editor and reviewers for their positive final assessment of our work. We have incorporated in the new version the few minor comments mentioned by Reviewer #1.

**Answers to the comments of Reviewer #1**

The manuscript contains some amendments compared to the previous version. The content is easily understandable and the writing is very good, so the material is straightforward reading. I am still not convinced of the usefulness of the two counterfactual scenarios; however, I am not raising the issue again and have just few remarks, most of which are minor stuff.

404: the third explanation given does not sound very convincing. Furthermore, the use of "slightly" does not seem very appropriate for the deviations seen in the plot.

We agree that the third explanation alone would not explain the differences between the observed and modelled water depths. It is probably the combination of the three factors that migth explain those differences.

Following the reviewer's suggestion, we have removed the word "slightly", and we have added the following sentence: "*These three factors might have contributed, to different degrees, to the deviations shown in Figure 11, the first two being probably the most relevant*".

492: the tone of this statement (using "not so accurately") sounds quite different from that used at line 408, where the model accuracy was declared as satisfactory in spite of the uncertainties involved. It is just an impression, but some more coherent writing could be used in the two places.

We appreciate the reviewer's impression, so we have modified, in line 404, the statement "*are satisfactory*" by "*follow the same trend as the observations*".

91: I would invert: "The PL mountain range is located in the ...".

This sentence was actually removed from the manuscript, since it was redundant. The location of the PL mountain range is already mentioned in the previous sentence, and also shown in Figure 1.

158-159: it is probably better to use "on" for days.

Modified.

337: is not

Modified.

**Answers to the comments of Reviewer #2**

After this round of revisions, I found a nicely improved paper. All remarks I raised, as a reviewer, were satisfactorily accomplished and\or replied.

We thank the reviewer for his positive assessment of the effort done in the revision of the manuscript.